# A Large-scale Training Paradigm for Graph Generative Models

**Yu Wang**[1]**, Ryan A. Rossi**[2]**, Namyong Park, Huiyuan Chen, Nesreen K. Ahmed**[3]**,
Puja Trivedi**[4]**, Franck Dernoncourt**[2]**, Danai Koutra**[5]**, Tyler Derr**[6]
[1]University of Oregon     [2]Adobe Research     [3]Cisco AI Research
[4]Amazon     [5]University of Michigan     [6]Vanderbilt University
yuwang@uoregon.edu, {ryrossi,dernonco}@adobe.com,
pujatriv@amazon.com, dkoutra@umich.edu, tyler.derr@vanderbilt.edu,
{park.namyong,n.kamel}@gmail.com, chenhuiyuan@gmail.com

## Abstract

Large Generative Models (LGMs) such as GPT, Stable Diffusion, Sora, and Suno are trained on a huge amount of texts, images, videos, and audio that are extremely diverse from numerous domains. This large-scale training paradigm on diverse well-curated data enhances the creativity and diversity of the generated content. However, all previous graph-generative models (e.g., GraphRNN, MD-VAE, MoFlow, GDSS, and DiGress) have been trained only on one dataset each time, which cannot replicate the revolutionary success achieved by LGMs in other fields. To remedy this crucial gap, we propose a large-scale training paradigm that uses a large corpus of graphs (over 5000 graphs) from 13 domains, leading to the development of LARGE GRAPH GENERATIVE MODELS (LGGMs). We empirically demonstrate that the pre-trained LGGMs have superior zero-shot generative capability to existing graph generative models. Furthermore, our pre-trained LGGMs can be easily fine-tuned with graphs from target domains and demonstrate even better performance than those directly trained from scratch, behaving as a solid starting point for real-world customization. Inspired by Stable Diffusion, we further equip LGGMs with the Text-to-Graph generation capability, such as providing the description of the network name and domain (i.e., "The power-1138-bus graph represents a network of buses in a power distribution system.") and network statistics (i.e., "The graph has a low average degree, suitable for modeling social media interactions."). This Text-to-Graph capability integrates the extensive world knowledge in the underlying language model, offering users fine-grained control of the generated graphs. We release the code, the model checkpoint, and the datasets at https://github.com/KINDLab-Fly/LGGM.

## 1 Introduction

Recently, Large Generative Models (LGMs) such as GPT, Stable Diffusion, and Sora (Achiam et al., 2023; Brooks et al., 2024) have achieved revolutionary success in generating creative and diverse content, which significantly increases the productivity of real-world applications (Somepalli et al., 2023). Unlike previous models such as Bert (Devlin et al., 2018) in Natural Language Processing (NLP) and Unet (Ronneberger et al., 2015) in image segmentation that are trained only on small-scale datasets from specific domains over narrow tasks, the key to the success of these LGMs lies in their large-scale training paradigm over well-curated training data from a wide variety of domains (Bubeck et al., 2023). Graph, as a different data modality from image, text, and audio, is ubiquitous in numerous fields and presents a new frontier for the applications of generative models such as drug discovery (Liu et al., 2023; Igashov et al., 2024; Liu et al.), material design (Liu et al., 2024a) and cyber-security (Liu et al., 2024c). Given the unprecedented success achieved by LGMs in other domains and the promising practical usage of graph generative models, we naturally ask:

*Can we design a large-scale training paradigm for graph generative models?*

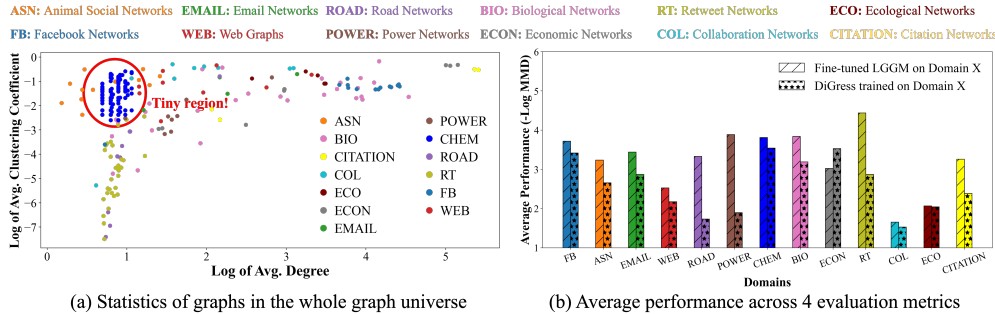

(a) Statistics of graphs in the whole graph universe   (b) Average performance across 4 evaluation metrics

Figure 1: (a): Average degree and clustering coefficient of graphs from 13 domains. The graph universe consists of graphs from distinct domains (e.g., the tiny region of Chemical Graphs), yet there are some common transferrable patterns. (b): Our pre-trained LGGM after fine-tuning on each domain achieves better generative performance than DiGress trained on that same domain.

Although graph generative models have been the long-standing focus of generative-based research (Zhu et al., 2022), previous ones have been trained on graphs from a single domain each time. For example, both the representation autoregressive-based GraphRNN (You et al., 2018), VAE-based GraphVAE (Simonovsky & Komodakis, 2018) and diffusion-based DiGress (Vignac et al., 2023) have been trained only on synthetic or chemistry graphs, the statistics of which only count a tiny region of the graph universe. In Figure 1(a), road networks have lower average clustering coefficients than Facebook networks because road networks have square intersections, while social relationships in Facebook networks form triangular connections (Rossi & Ahmed, 2019). Tortoises Animal Social Networks (Sah et al., 2019) have a lower average degree than Power Networks because tortoises, as solitary creatures, would not share the same burrow (Sah et al., 2016). Therefore, graph generative models trained by graphs from one domain can hardly generalize to unseen graphs, as shown by the worse zero-shot performance of DiGress in Table 2. Moreover, without training on graphs covering the graph universe, small models can never replicate the success achieved by LGMs in other fields.

Recognizing the significant gap in developing LGMs for graph-structured data and their potential revolutionary impact similar to LGMs in other fields, we design the very first large-scale training paradigm that leads to the development of LARGE GRAPH GENERATIVE MODELS (LGGMs) pre-trained over 5000 graphs from 13 domains sourcing from the Network Repository (Rossi & Ahmed, 2015; 2016). After pretraining, our LGGM learns fundamental structural patterns that are transferable across different domains (Mao et al., 2024) and henceforth demonstrates better zero-shot generation on graphs from unseen domains in Table 2. The generated graphs from pre-trained LGGMs using the same domain further boost the graph classification performance in Table 3. Moreover, the pre-trained LGGMs are highly adaptable for fine-tuning on a specific domain, achieving an overall performance increase of 30% compared to the smaller DiGress model trained on the same domain, as depicted in Figure 1(b). More importantly, our LGGMs support Text-to-Graph generation, which allows for finer-level control of the generated graphs (e.g., their domains/names in Table 4 and clustering coefficient/average degree in Figure 5). Our contributions are as follows:

- **Large Graph Generative Models:** We explore the large-scale training paradigm and propose Large Graph Generative Models (LGGMs), trained on thousands of graphs arising from 13 distinct domains. To the best of our knowledge, this work is the first to explore the potential of the large-scale training paradigm on graph-structured data. We hope others expand this collection and leverage our work to develop future LGGMs that could eventually replicate or exceed the success of Stable Diffusion (Rombach et al., 2022) but in the graph modality.

- **Superior Zero-shot and Fine-tuning Generative Capability:** Our pre-trained LGGM delivers exceptional zero-shot generative performance on unseen graphs in Table 2 of Section 5.2 and shows great adaptability for fine-tuning in Figure 3 of Section 5.3. Remarkably, the fine-tuned LGGM outperforms DiGress trained from scratch on the same graphs by around 30%, especially under limited data scenarios , behaving as a better starting point for real-world development.

- **Text-to-Graph Generation:** We equip the LGGM with the capability to generate graphs given user-specified text prompts, allowing finer-level control of the generated graphs in terms of their domains/names and network statistics such as degree and clustering coefficient.

## 2 RELATED WORK

### 2.1 LARGE GENERATIVE MODELS (LGMs)

Recent years have witnessed unprecedented success achieved by LGMs (Achiam et al., 2023; Brooks et al., 2024; Cao et al., 2024; Touvron et al., 2023; Zhou et al., 2023). For example, in natural language processing (NLP), large language models can effectively produce human-readable texts for tasks such as question answering, translation, and more (Qin et al., 2023a; Tian et al., 2024; Han et al., 2024). Furthermore, multi-modal generative models now support cross-modality generation, such as converting text into images or vice versa (Rombach et al., 2022; Zhang et al., 2023; Wang et al., 2022a). The key to their success lies in their usage of the world knowledge inherited from the pre-training stage. This world knowledge has demonstrated positive transferability across numerous domains. Compared with the recent LGMs in NLP/computer vision (Liu et al., 2024b; Penedo et al., 2023; Rombach et al., 2022; Touvron et al., 2023), we alternatively develop LGMs for graphs to realize a similar set of advantages achieved by LGMs in other fields, including enhanced zero-shot generalization, improved fine-tuning performance and cross-modality generation.

### 2.2 GRAPH GENERATIVE MODELS

Given the ubiquity of graphs in modeling relational information of real-world objects across different domains (Rossi & Ahmed, 2015; Hu et al., 2020; Liu et al., 2024d; Wang et al., 2024; Li et al., 2023), graph generative models have been developed to generate realistic graphs for advancing numerous applications (Kang et al., 2024; Liu et al., 2024a; Trivedi et al., 2024), such as generating molecules with high drug-likeness, designing imperceptible adversarial attacks, and supporting conditional gener-

Table 1: Comparison between previous graph generative models and our proposed LGGMs.

| Type | Model | # Domains | Multi-Domain Training | Text2Graph |
|------|-------|-----------|------------------------|------------|
| AR | (You et al., 2018) | 2 | ✘ | ✘ |
| | (Bacciu et al., 2020) | 3 | ✘ | ✘ |
| VAE | (Du et al., 2022) | 1 | ✘ | ✘ |
| | (Guo et al., 2021) | 1 | ✘ | ✘ |
| GAN | (Maziarka et al., 2020) | 1 | ✘ | ✘ |
| | (Fan & Huang, 2019) | 2 | ✘ | ✘ |
| FLOW | (Madhawa et al., 2019) | 1 | ✘ | ✘ |
| | (Zang & Wang, 2020) | 1 | ✘ | ✘ |
| | (Luo et al., 2021) | 2 | ✘ | ✘ |
| DIFF | (Jo et al., 2022) | 3 | ✘ | ✘ |
| | (Vignac et al., 2023) | 2 | ✘ | ✘ |
| | (Liu et al., 2021) | 1 | ✘ | ✘ |
| **LGGMs - Ours** | | **13** | ✔ | ✔ |

ation. Graph generative models can generally be divided into two categories: statistic-based ones and deep learning-based ones. Statistic-based generative models such as Stochastic Block Models (Lee & Wilkinson, 2019) and Small World Models (Newman, 2000) assume that the real-world graph formation adheres to specific statistical rules, and define various sampling strategies to simulate networks with prescribed properties. However, this approach oversimplifies the complex distribution of real graphs and struggles to generalize to those deviating from established norms. This limitation has spurred recent research into deep-learning-based generative models that automatically capture intricate statistics by learning to recover graphs (Simonovsky & Komodakis, 2018; Vignac et al., 2023; Trivedi et al., 2024; You et al., 2018; Zang & Wang, 2020). Despite their effectiveness, they all focus on a narrow range of domains and are trained solely on a single domain each time, as shown in Table 1. Instead, we focus on training graph generative models in a large-scale paradigm with thousands of graphs from 13 domains.

## 3 LARGE-SCALE TRAINING PARADIGM OF LGGM

### 3.1 NOTATION

Let $\mathbb{G}$ be a random variable of universal graphs, governed by its underlying distribution $P(\mathbb{G})$. Given that real-world graphs originate from various domains, we introduce $\mathbb{G}^c$ to represent a random variable for graphs from domain $c$, with its distribution as $P(\mathbb{G}^c)$. Assuming the universal graph space encompasses $\mathcal{C}$ distinct domains, i.e., $\mathcal{G} = \cup_{c \in \mathcal{C}} \mathcal{G}^c$ with each set of graphs from domain $c$ as $\mathcal{G}^c$, then $P(\mathbb{G}^c)/P(\mathbb{G})$ is domain-specific/agnostic distribution. To ease the introduction of training and evaluation setting in Section 5, we further divide each domain-specific set of graphs $\mathcal{G}^c$ into training, validation and testing subsets, notated as $\mathcal{G}^c = \mathcal{G}^{\text{Train},c} \cup \mathcal{G}^{\text{Val},c} \cup \mathcal{G}^{\text{Test},c}$. We represent each graph $G = (\mathbf{X}^G, \mathbf{E}^G)$ with $\mathbf{X}^G \in \mathbb{R}^{n_G \times d_X}/\mathbf{E}^G \in \mathbb{R}^{n_G \times n_G \times d_E}$ as the one-hot encoding matrix representing node/edge categories with $n_G$ being the number of nodes in graph $G$ and $d_X/d_E$ being the number of node/edge categories, considering the edge existence as a particular edge category. In Text-to-Graph generation, each graph $G$ is paired with a textual description $S$ from the textual distribution $P(\mathbb{S})$ and their joint distribution is $P(\mathbb{G}, \mathbb{S})$.

## 3.2 LARGE GRAPH CORPUS

Training LGGM requires a substantial, well-curated collection of graphs from multiple domains. We select graphs from the Network Repository across 13 distinct yet representative domains covering a wide variety of real-world scenarios, including Facebook (FB), Animal Social (ASN), Email, Web, Road, Power, Chemical (CHEM), Biological (BIO), Economic (ECON), Retweet (RT), Collaboration (COL), Ecological (ECO), Citation, as shown in Figure 2(a). Given that many real-world graphs (e.g., social networks and road networks) comprise thousands or even millions of nodes and edges, and that state-of-the-art diffusion models, e.g., DiGress and GDSS, are limited to handling networks with only hundreds of nodes, we further sample subgraphs for certain domains to address scalability challenges. Specifically, we generate 2- and 3-hop ego subgraphs centered on multiple randomly chosen nodes followed by taking their induced subgraphs (Trivedi et al., 2024; Limnios et al., 2023). We apply this strategy iteratively across all the initially collected graphs until hitting the preset budget. Appendix D.1 presents the graph statistics.

## 3.3 PRE-TRAINING AND GRAPH GENERATION OF LGGMs

Our LGGMs are designed based on discrete denoising diffusion (Chen et al., 2023; Vignac et al., 2023), which consists of a forward process based on a transition matrix and a reverse process based on minimizing the cross-entropy loss between the ground-truth and predicted clean graphs.

During the forward process, for each graph $G$ sampled from the distribution $P(\mathbb{G})$, we obtain its noisy version $G^t = (\mathbf{X}^t, \mathbf{E}^t)$ at step $t$ by sampling from the conditional categorical distribution: $q(\mathbb{G}^t|\mathbb{G}^{t-1}) = (\mathbf{X}^{t-1}\mathbf{Q}_X^t, \mathbf{E}^{t-1}\mathbf{Q}_E^t)$ and $q(\mathbb{G}^t|\mathbb{G}^0) = (\mathbf{X}\bar{\mathbf{Q}}_X^t, \mathbf{E}\bar{\mathbf{Q}}_E^t)$ where $\mathbf{Q}_X^t \in \mathbb{R}^{d_X \times d_X}$ and $\mathbf{Q}_E^t \in \mathbb{R}^{d_E \times d_E}$ are node/edge transition matrices and $\mathbb{G}^0 = \mathbb{G}$ is the original data distribution of graphs. Depending on whether our generative downstream tasks require generalization to unseen domains or not, we can either use different transition matrices for graphs from different domains, i.e., domain-specific transition matrix $\mathbf{Q}_X^{t,c} = \alpha^t\mathbf{I} + (1 - \alpha^t)\mathbf{1}\mathbf{m}_X^c, \mathbf{m}_X^c = \frac{1}{|\mathcal{G}^{\text{Train},c}|}\sum_{G \in \mathcal{G}^{\text{Train},c}}\mathbf{X}^G, \forall c \in C$ or unify transition matrices across different domains. For the unified transition matrices, we can trivially use the uniform transition matrix, i.e. $\mathbf{Q}_X^{t,c} = \alpha^t\mathbf{I} + (1 - \alpha^t)(\mathbf{1}_{d_X}\mathbf{1}_{d_X}^\top)/d_X$, or compute the marginal transition matrix across all graphs from all domains $\mathbf{Q}_X^t = \alpha^t\mathbf{I} + (1 - \alpha^t)\mathbf{1}\mathbf{m}_X, \mathbf{m}_X = \frac{1}{|\mathcal{G}^{\text{Train}}|}\sum_{G \in \mathcal{G}^{\text{Train}}}\mathbf{X}^G$. And $\mathbf{Q}_E^t$ can be computed similarly. We validate the advantages of LGGMs under both of these two transition strategies in Appendix F.

In the reverse process, a parametrized neural network is trained to predict the clean graph given the sampled noisy graph by optimizing the following loss:

$$\mathbf{\Theta}^\star = \arg\min_{\mathbf{\Theta}} \mathcal{L} = \mathbb{E}_{G \sim P(\mathbb{G})}\mathbb{E}_{t \sim \mathcal{T}}\mathbb{E}_{G^t \sim q(\mathbb{G}^t|\mathbb{G})}(-\log p_{\mathbf{\Theta}}(G|G^t)). \tag{1}$$

Following Vignac et al. (2023), we combine the learned $P_{\mathbf{\Theta}^\star}(\mathbb{G}|\mathbb{G}^t)$ and the closed-form posterior $P(\mathbb{G}^{t-1}|\mathbb{G}^t, \mathbb{G})$ to perform backward generation by sampling from the following distribution:

$$P(\mathbb{G}^{t-1}|\mathbb{G}^t) \propto \sum_{\mathbb{G}} P(\mathbb{G}^{t-1}|\mathbb{G}^t, \mathbb{G})P_{\mathbf{\Theta}^\star}(\mathbb{G}|\mathbb{G}^t). \tag{2}$$

## 3.4 FINE-TUNING LGGMs

In many real-world applications, the graphs of interest $\widetilde{\mathcal{G}}$ may highly likely come from completely unseen domains, i.e., $\widetilde{\mathcal{G}} \cap \mathcal{G} = \emptyset$, and their corresponding distribution may also be significantly different from the pre-trained one, i.e., $P(\widetilde{\mathbb{G}}) \neq P(\mathbb{G})$ as shown by comparing CHEM and FB Networks in Figure 1(a). In this case, we further fine-tune our pre-trained LGGMs based on the observed graphs $\widetilde{\mathcal{G}}$ from the unseen domains:

$$\mathbf{\Theta}^{\star\star} = \arg\min_{\mathbf{\Theta}} \mathcal{L} = \mathbb{E}_{\widetilde{G} \sim P(\widetilde{\mathbb{G}})}\mathbb{E}_{t \sim \mathcal{T}}\mathbb{E}_{\widetilde{G}^t \sim q(\widetilde{\mathbb{G}}^t|\widetilde{\mathbb{G}})}(-\log p_{\mathbf{\Theta}}(\widetilde{G}|\widetilde{G}^t)), \tag{3}$$

where $\mathbf{\Theta}^{\star\star}$ is initialized as $\mathbf{\Theta}^\star$ from the pretaining phase in Eq. (1). After fine-tuning, our LGGM can effectively adapt to unseen distributions by using both the prior knowledge from the pre-training stage and the specific knowledge of new graphs from the unseen domains, as verified in Figure 3.

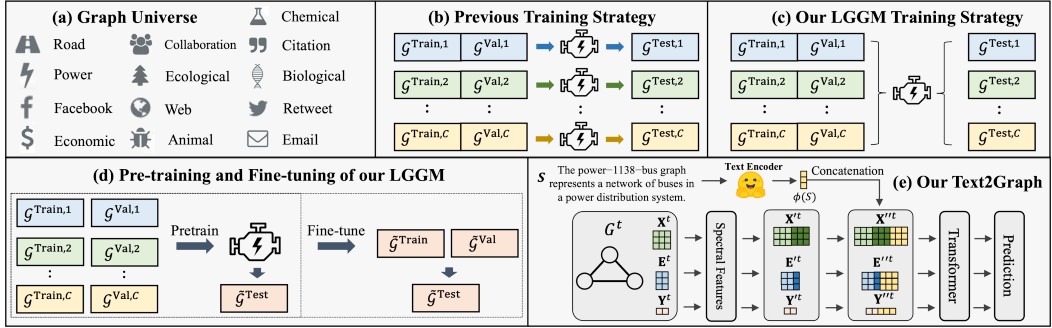

Figure 2: The overview of LGGM framework and experimental settings. (a): Graph universe includes our collected 13 distinct yet representative domains. (b)-(c): Compared with all previous graph generative models that have been trained only on one domain each time, our LGGMs are trained on thousands of graphs from 13 domains. (d): We pre-train/fine-tune LGGMs in Section 3.3/3.4. (e): Given the text prompt $S$ and the current generated graph at $t$, we concatenate its textual embedding obtained from a pre-trained language model with the node/edge/graph embeddings after spectral feature extraction, and forward them through the Graph Transformer to predict the clean graph.

Table 6 presents the Time/Space complexity of LGGMs in Appendix B. The proposed pre-trained and fine-tuned LGGMs mimic the random sampling from the learned distribution $P(\mathbb{G})$ that is prescribed by the training data without any fine-level customization. To control the characteristics of the generated graphs, we further propose Text-to-Graph LGGMs to generate graphs based on textual descriptions. In this way, users could specify properties of the graphs through natural language description, thereby customizing the graph generation.

## 4 TEXT-TO-GRAPH LGGM

Given the textual description $S$ about the network to be generated, our goal here is to learn $P(\mathbb{G}^{t-1}|\mathbb{G}^t, \mathbb{S})$, which is further decomposed as:

$$P(\mathbb{G}^{t-1}|\mathbb{G}^t, \mathbb{S}) \propto \sum_{\mathbb{G}} P(\mathbb{G}^{t-1}|\mathbb{G}^t, \mathbb{G}, \mathbb{S})P(\mathbb{G}|\mathbb{G}^t, \mathbb{S}). \tag{4}$$

Theorem 1 proves that if the transition matrices $\mathbf{Q}_X^t, \mathbf{Q}_E^t$ in forward process are independent of the textual description $S$, the first term $P(\mathbb{G}^{t-1}|\mathbb{G}^t, \mathbb{G}, \mathbb{S})$ can then be simplified as $P(\mathbb{G}^{t-1}|\mathbb{G}^t, \mathbb{G})$ with the analytical form computation (Vignac et al., 2023). For the second term, we approximate it by a neural network, i.e., $P(\mathbb{G}|\mathbb{G}^t, \mathbb{S}) = P_{\mathbf{\Theta}'}(\mathbb{G}|\mathbb{G}^t, \mathbb{S})$ with $\mathbf{\Theta}'$ being obtained by:

$$\mathbf{\Theta}' = \arg\min_{\mathbf{\Theta}} \mathcal{L} = \mathbb{E}_{(G,S)\sim P(\mathbb{G},\mathbb{S})}\mathbb{E}_{t\sim\mathcal{T}}\mathbb{E}_{G^t\sim q(\mathbb{G}^t|\mathbb{G})}(-\log p_{\mathbf{\Theta}}(G|G^t, \phi(S))), \tag{5}$$

where $\phi$ is a pre-trained textual encoder. Figure 2(e) shows the architecture of LGGM-Text2Graph, which firstly integrates the textual embedding $\phi(S)$ into the node/edge/graph-level latent embeddings after spectral feature extraction of the current generated graph and further predicts the clean graph. Theorem 2 proves that modeling $P(\mathbb{G}^{t-1}|\mathbb{G}^t, \mathbb{S})$ with $P_{\mathbf{\Theta}'}(\mathbb{G}^{t-1}|\mathbb{G}^t, \mathbb{S})$ leads to higher evidence lower bound of the likelihood $\log P(\mathbb{G}^0, \mathbb{S})$.

Training $p_{\mathbf{\Theta}}(G|G^t, \phi(S))$ in Eq. (5) requires the joint distribution between graphs and their corresponding textual descriptions, i.e., $P(\mathbb{G}, \mathbb{S})$. Given users' specific interests in the graphs to generate, we explore two main categories of textual prompts to guide graph generation: domain/name (e.g., Power Network, power-1138-bus) and structural characteristics (e.g., average degree, clustering coefficient). For example, zoologists interested in the dynamics of tortoise interactions might seek to generate Animal Social Networks (Sosa et al., 2021), and social scientists studying social anomalies might prioritize generating social interactions with dense and unexpected connections (Ma et al., 2021; Akoglu et al., 2015). Since this work is a pioneering effort in Text-to-Graph generation and no prior collection of user prompts for this purpose exists, following previous works, e.g., LLaVA (Liu et al., 2024b; Zhang et al., 2023), we ask GPT3.5/4 to emulate the human drafting of prompts to obtain

pairs of (user prompt, graph). To prepare the graphs with user prompts about their domains/names, we obtain the domain/name information of each graph directly from the Network Repository and prompt GPT3.5 to generate the human-readable description paired with the corresponding graph. See more details in Appendix D.2. For preparing the graphs with user prompts about their average clustering coefficient/degree, instead of using graphs from Network Repository that only count partially of the entire graph universe (i.e., no existing graphs there cover the area with high average degree and low average clustering coefficient in Figure 1(a)), we use the Watts–Strogatz model to synthesize graphs covering the full spectrum of the graph universe. After that, we calculate the average degree and clustering coefficient for each graph and prompt GPT4 to generate textual descriptions about these networks using their statistics. See more details in Appendix D.3. We also employ t-SNE visualization to analyze the generated textual descriptions, as shown in Figure 7. This visualization indicates that texts describing graphs from various domains or with distinct statistics tend to form separate clusters, a necessary condition for the successful control of the generated graphs.

## 5 EXPERIMENTS

### 5.1 EXPERIMENTAL SETUP

In this section, we conduct five experiments over the graphs collected from 13 domains to demonstrate the effectiveness of LGGMs in five different aspects, the details of which are summarized as follows:

- **Pre-training Evaluation in Table 2 in Section 5.2:** To demonstrate the superior zero-shot performance of LGGM in generating unseen graphs compared to conventional graph generative models, we adopt the out-of-distribution evaluation where we iteratively treat each domain X as the unseen one and train the LGGM using training graphs from all other domains, and evaluate its performance on the testing graphs from the unseen domain X. The variant of LGGM in this experiment is called LGGM-X where X represents the unseen domain.

- **Fine-tuning Evaluation in Figure 3 in Section 5.3:** To demonstrate the high adaptability for fine-tuning LGGM, we further fine-tune the above pre-trained LGGM. Specifically, we take LGGM-X pre-trained on graphs from all other domains but domain X, and then fine-tune it on the training graphs from domain X. After that we evaluate it on the testing graphs from domain X. The variant of LGGM in this experiment is called Fine-tuned LGGM on X.

- **Fine-tuned LGGM compared with DiGress trained directly on X in Figure 1(b)/4(c) in Section 5.4:** When having access to graphs of domain X, users could directly train existing graph generative models and generate graphs for the domain X. To demonstrate the practical usage of LGGMs, we further compare the fine-tuned LGGM on X with DiGress/EDGE directly trained on X. In addition, we also compare their performance under limited data scenarios (Gavrilev & Burnaev, 2023; Liu et al., 2024a) in Figure 4(a)-(b).

- **Graph classification after boosting the training data with our Pre-trained LGGM compared with using original training data in Table 3 in Section 5.4:** When the training data for discriminative tasks like graph classification is insufficient, users could use our pre-trained LGGM to generate graphs and boost the training data. To demonstrate the benefits of incorporating these generated graphs during the training process, we compare graph classification performance before and after we boost the training data with generated graphs by our LGGM pre-trained on graphs from chemistry and social domains.

- **Text-to-Graph Generation in Table 4 and Figure 5 in Section 5.5:** To control the graph generation, we consider two types of user prompt information: the domain/name and the graph properties, i.e., we train LGGM on training graphs from all domains with user prompts either describing the graph domains/names or graph statistics. We call these two variants of LGGM as LGGM-T2G$^D$ and LGGM-T2G$^{UP}$, respectively.

Figure 8 in Appendix E.3 comprehensively illustrates each of the above training paradigms. Due to the page limitation, we present the evaluation metrics and model hyperparameters in Appendix E. Moreover, we only present results under the uniform transition strategy in the main paper while leaving the one under domain-specific transition strategy in Appendix F. It is important to note that the benefits of LGGMs are consistent across both of these two transition strategies.

Table 2: Comparing Zero-shot Generative Performance on unseen Graphs in held-out domain X between DiGress trained on QM9 and LGGM-X trained on all except the held-out domain X. Result "ALL" is computed by averaging across 12 domains and the best result for each domain is in **bold**.

| Domain | Method | DEG | CC | Spec | Orb | Domain | Method | DEG | CC | Spec | Orb |
|---|---|---|---|---|---|---|---|---|---|---|---|
| FB | DiGress | **0.3376** | **0.6298** | **0.0797** | **0.3593** | BIO | DiGress | 0.2712 | 0.5202 | 0.1127 | 0.3188 |
|  | LGGM-X | 0.4723 | 0.6843 | 0.2924 | 0.7555 |  | LGGM-X | **0.1081** | **0.2696** | **0.0900** | **0.2053** |
| ASN | DiGress | 0.1496 | 0.3258 | 0.1506 | 0.4420 | ECON | DiGress | 0.2987 | 0.4841 | 0.2162 | 0.3834 |
|  | LGGM-X | **0.0281** | **0.2440** | **0.0830** | **0.0618** |  | LGGM-X | **0.1213** | **0.0920** | **0.1120** | **0.1086** |
| EMAIL | DiGress | 0.2192 | 0.6012 | **0.0702** | 0.3416 | RT | DiGress | 0.4164 | **0.1327** | 0.4147 | 0.5957 |
|  | LGGM-X | **0.0751** | **0.2364** | 0.0768 | **0.3089** |  | LGGM-X | **0.0525** | 0.1429 | **0.1330** | **0.2219** |
| WEB | DiGress | 0.2556 | 0.6186 | 0.1877 | 0.6045 | COL | DiGress | 0.2473 | 0.5826 | 0.2314 | 0.7679 |
|  | LGGM-X | **0.0648** | **0.3961** | **0.0549** | **0.1127** |  | LGGM-X | **0.0736** | **0.5769** | **0.0895** | **0.0988** |
| ROAD | DiGress | 0.3705 | 0.8226 | 0.2801 | 0.7198 | ECO | DiGress | 0.5431 | 0.7915 | **0.2338** | 0.6045 |
|  | LGGM-X | **0.0713** | **0.2193** | **0.0987** | **0.2986** |  | LGGM-X | **0.4753** | **0.3904** | 0.3194 | **0.3934** |
| POWER | DiGress | 0.3726 | 0.4582 | 0.3270 | 1.4732 | CITATION | DiGress | 0.2527 | 0.7790 | 0.1315 | **0.4966** |
|  | LGGM-X | **0.0119** | **0.1293** | **0.0373** | **0.0754** |  | LGGM-X | **0.1348** | **0.7257** | **0.1160** | 0.4981 |
| ALL | DiGress | 0.3112 | 0.5622 | 0.2030 | 0.5923 |  |  |  |  |  |  |
|  | LGGM-X | **0.1408** | **0.3422** | **0.1253** | **0.2616** |  |  |  |  |  |  |

**DEG**, **CC**, **Spec**, **Orb**: MMD of Degree, Clustering Coefficient, Eigenvalues, and Orbits, more details are in Appendix E.1.

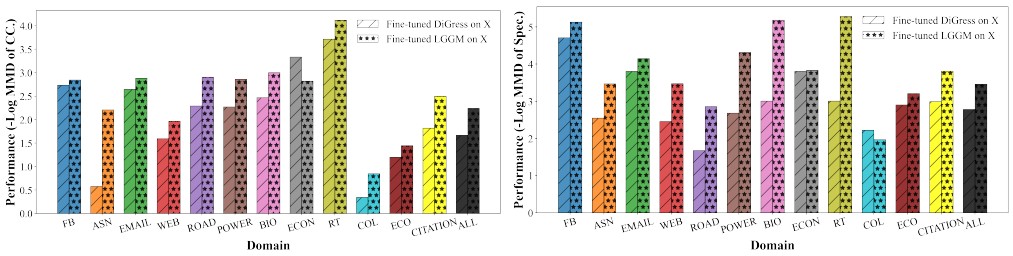

(a) Performance of MMD of CC.  (b) Performance of MMD of Spec.

Figure 3: Performance comparison between Fine-tuned LGGM and Fine-tuned DiGress.

## 5.2 PRE-TRAINING EVALUATION

Table 2 compares the performance of our model, LGGM-X, pre-trained on all graph domains except the held-out domain X, with DiGress trained on the QM9 dataset. Both of them are evaluated over graphs from the unseen domain X. Overall, LGGM-X outperforms DiGress across all evaluation metrics shown by the "ALL" result. This superiority suggests that training on graphs from diverse domains captures transferable structural patterns and enhances the generalization of the model to unseen domains. The only exception from this trend occurs with Facebook Networks (FB) where our LGGM-X performs uniformly worse than DiGress across all evaluation metrics. This is because Facebook Networks (FB) only count a tiny region among the whole graph universe. As illustrated in Figure 1(a), the average clustering coefficient of FB graphs ranges from 0.301 to 0.407, a narrow segment within the broader global graph spectrum spanning from 0 to 1. This narrow range poses a challenge for the generalized LGGM-X to specialize in learning the graph data distribution specific to the FB domain. Furthermore, we conduct the same experiment but under the domain-specific transition strategy in Table 8 in Appendix F, and similarly, LGGM-X generally outperforms DiGress.

## 5.3 FINE-TUNING EVALUATION

In addition to the superior zero-shot generative performance of pre-trained LGGM-X, many real-world applications already possess exemplary graphs that can be leveraged, e.g., different types of anomaly behaviors in social networks/e-commerce platforms, and molecules with predefined chemical structures in drug discovery. In these scenarios, users can fine-tune LGGM-X with these domain-specific graphs, adapting the broadly trained model to specialize in generating graphs tailored to target domains. Figure 3 compares the generative performance of fine-tuned DiGress on X that is originally pre-trained on QM9 and fine-tuned LGGM-X on X that is originally pre-trained on all but domain X. We can see that LGGM-X consistently outperforms DiGress for graphs from most of the domains, which further validates the adaptability of LGGM after fine-tuning on a specific domain.

Table 3: Augmenting Graph Classification by Generating Graphs with LGGM. We follow the conventional 10 cross-validation setting for evaluation. Better performance is highlighted red , while worse performance is highlighted blue .

| Model | Strategy | Protein | | ENZYMES | | MUTAG | | IMDB-M | |
|---|---|---|---|---|---|---|---|---|---|
| | | F1-macro | F1-micro | F1-macro | F1-micro | F1-macro | F1-micro | F1-macro | F1-micro |
| GIN | Basic | 0.727±0.04 | 0.739±0.04 | 0.614±0.09 | 0.680±0.08 | 0.810±0.15 | 0.831±0.13 | 0.449±0.04 | 0.489±0.04 |
| | LGGM$^{Chem}$ | 0.735±0.03 | 0.748±0.03 | 0.615±0.08 | 0.693±0.05 | 0.766±0.18 | 0.799±0.14 | 0.482±0.03 | 0.487±0.04 |
| | LGGM$^{Soc}$ | 0.724±0.04 | 0.736±0.04 | 0.597±0.08 | 0.697±0.04 | 0.839±0.09 | 0.861±0.08 | 0.483±0.04 | 0.493±0.04 |
| GCN | Basic | 0.688±0.04 | 0.706±0.04 | 0.631±0.04 | 0.690±0.03 | 0.747±0.10 | 0.782±0.09 | 0.468±0.06 | 0.498±0.06 |
| | LGGM$^{Chem}$ | 0.724±0.02 | 0.736±0.02 | 0.615±0.08 | 0.693±0.06 | 0.840±0.12 | 0.851±0.12 | 0.480±0.03 | 0.486±0.05 |
| | LGGM$^{Soc}$ | 0.727±0.05 | 0.739±0.05 | 0.611±0.11 | 0.690±0.05 | 0.795±0.14 | 0.814±0.13 | 0.464±0.06 | 0.483±0.05 |

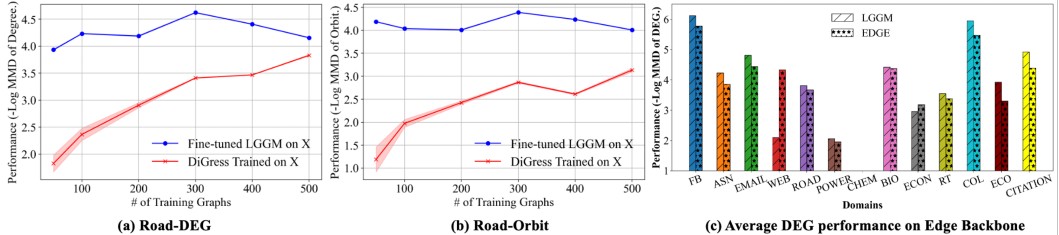

Figure 4: (a)-(b): With fewer training graphs, Fine-tuned LGGM becomes more advantageous than DiGress. More analysis is in Appendix F.7. (c) We further demonstrate our large-scale training paradigm gains similar advantages when equipped with the graph generative backbone EDGE (Chen et al., 2023), complete results of which are in Appendix F.5.

## 5.4 PRACTICAL USAGE OF FINE-TUNED LGGM

To demonstrate the practical usage of LGGM in generating graphs for real-world deployment, we further compare the fine-tuned LGGM with DiGress trained directly on each domain in Figure 1(b). We can see that even using the same graphs for training, due to the additional knowledge incorporated during the pre-training phase of LGGM, it exhibits significantly better generative performance for most domains. Moreover, this advantage becomes even more pronounced when fewer graphs are available. Figure 4(a)-(b) illustrates the improved performance of the fine-tuned LGGM versus DiGress trained in X, with a wide margin as the number of training graphs in X decreases. This is particularly useful since many graph-generative applications involve semi-supervised settings, e.g., generating anomaly software and drug design, the amount of which only count 0.05%-0.5% (Bajorath, 2002) and 0.01% (Oak et al., 2019) among the whole potential candidates, respectively. We further demonstrate that the advantages of our proposed large-scale training paradigm are not limited to DiGress but can also generalize to other generative backbones. In Figure 4(c), we apply our large-scale training paradigm to EDGE and compare the graph generative performance. We observe that LGGM also achieves better performance than EDGE. The smaller performance margin after switching to the EDGE backbone, we hypothesize, is due to EDGE essentially approximating the backward process by only recovering edges incident to important nodes. Therefore, training EDGE with many graphs from different domains may cause less accurate crucial node estimation.

In addition to enhancing generative tasks, the generated graphs can also augment the training data and improve graph classification. We select TUDataset (Morris et al., 2020), a well-established graph classification dataset consisting of graphs from chemistry and social domains. For each graph in the training dataset, we use LGGM to diffuse and regenerate its adjacency matrix while keeping the node features initialized as in the original graph. The generated graphs are then added to the training set, and the graph classification model is retrained with the boosted dataset. As shown in Table 3, the graph classification performance improves in most cases when we use LGGM to augment the training graphs. More specifically, F1-macro usually achieves a larger improvement than F1-micro. This is because F1-macro gives more weight to minority classes, and hence the boosting benefit is naturally more pronounced on minority data rather than majority data. Interestingly, we also observe that this performance benefit sometimes occurs across different domains; for example, generating graphs with LGGM trained on the social domain could lead to better performance on MUTAG. This sheds light on domain transferability (Mao et al., 2024), where we conduct initial analysis in Appendix F.4 but leave deeper exploration as future work.

Table 4: Comparing the Graph Generative Performance of LGGM with/without Text Conditions. Best and runner-up results are **bolded** and underlined.

| Domain | Method | DEG | CC | Spec | Orb | Domain | Method | DEG | CC | Spec | Orb |
|---|---|---|---|---|---|---|---|---|---|---|---|
| FB | LGGM | 0.0321 | 0.4994 | 0.0763 | 0.3117 | BIO | LGGM | 0.2661 | 0.3120 | 0.1135 | 0.3835 |
| | LGGM-T2G$^D$ | 0.1561 | 0.1639 | 0.0924 | 0.0417 | | LGGM-T2G$^D$ | 0.0099 | 0.1286 | 0.0303 | 0.1366 |
| | LGGM-T2G$^{UP}$ | **0.0050** | **0.0545** | **0.0070** | **0.0251** | | LGGM-T2G$^{UP}$ | **0.0028** | **0.0287** | **0.0236** | **0.0174** |
| ASN | LGGM | 0.1511 | 0.4325 | 0.1875 | 0.3896 | ECON | LGGM | 0.3828 | 0.1533 | 0.2039 | 0.2583 |
| | LGGM-T2G$^D$ | 0.0318 | 0.2821 | 0.0606 | 0.0631 | | LGGM-T2G$^D$ | 0.0666 | 0.0594 | 0.0650 | 0.0586 |
| | LGGM-T2G$^{UP}$ | **0.0211** | **0.1191** | **0.0462** | **0.0195** | | LGGM-T2G$^{UP}$ | **0.0132** | **0.0257** | **0.0053** | **0.0191** |
| EMAIL | LGGM | 0.2156 | 0.2450 | 0.0666 | 0.2757 | RT | LGGM | 0.4395 | 0.2225 | 0.4337 | 0.6641 |
| | LGGM-T2G$^D$ | 0.0469 | 0.0982 | 0.0484 | 0.0505 | | LGGM-T2G$^D$ | 0.0468 | 0.0955 | 0.0729 | 0.0393 |
| | LGGM-T2G$^{UP}$ | **0.0073** | **0.0379** | **0.0127** | **0.0437** | | LGGM-T2G$^{UP}$ | **0.0286** | **0.0933** | **0.0400** | **0.0312** |
| WEB | LGGM | 0.2725 | 0.2672 | 0.1900 | 0.4368 | COL | LGGM | 0.3565 | 0.3554 | 0.2451 | 0.7874 |
| | LGGM-T2G$^D$ | 0.0255 | **0.0737** | 0.0354 | 0.1856 | | LGGM-T2G$^D$ | 0.0395 | 0.3110 | 0.1146 | 0.1823 |
| | LGGM-T2G$^{UP}$ | **0.0105** | 0.0941 | **0.0206** | **0.0451** | | LGGM-T2G$^{UP}$ | **0.0265** | **0.2813** | **0.0895** | **0.0899** |
| ROAD | LGGM | 0.4825 | 0.5373 | 0.3398 | 0.7542 | ECO | LGGM | 0.5466 | 0.6003 | 0.2257 | 0.7089 |
| | LGGM-T2G$^D$ | **0.0088** | 0.1225 | 0.0399 | 0.0155 | | LGGM-T2G$^D$ | 0.2160 | 0.2917 | 0.1203 | 0.2569 |
| | LGGM-T2G$^{UP}$ | 0.0177 | **0.0437** | **0.0336** | **0.0086** | | LGGM-T2G$^{UP}$ | **0.0293** | **0.2885** | **0.0416** | **0.2556** |
| POWER | LGGM | 0.4394 | 0.4646 | 0.3473 | 1.3186 | CITATION | LGGM | 0.2624 | 0.5374 | 0.1295 | 0.3419 |
| | LGGM-T2G$^D$ | 0.0162 | 0.1131 | 0.0479 | 0.1786 | | LGGM-T2G$^D$ | 0.0101 | 0.1025 | 0.0315 | 0.0651 |
| | LGGM-T2G$^{UP}$ | **0.0062** | **0.0570** | **0.0111** | **0.0084** | | LGGM-T2G$^{UP}$ | **0.0072** | **0.0849** | **0.0115** | **0.0287** |

(a) Average Clustering Coefficient

(b) Average Degree

Figure 5: Text-to-Graph Generation with Prescribed Graph Properties. (a) Controlling Average Clustering Coefficient; (b) Controlling Average Degree. GT-Ground Truth Graphs and Gen-Generated Graphs. Below each graph, the number of nodes and key statistical measures are displayed.

## 5.5 TEXT-TO-GRAPH GENERATION

Here we integrate Text-to-Graph (T2G) generation into LGGMs. We introduce two variants: LGGM-T2G$^D$, which utilizes domain labels such as "Power Networks" as textual descriptions, and LGGM-T2G$^{UP}$, which utilizes user prompts from GPT3.5, like "The power-1138-bus graph represents a network of buses in a power distribution system". Table 4 compares the basic LGGM trained without text conditions, against LGGM-T2G$^D$ and LGGM-T2G$^{UP}$. Firstly, we observe a significant performance improvement from LGGM to LGGM-T2G$^D$/LGGM-T2G$^{UP}$. The inclusion of text descriptions acts as a unique identifier that enables LGGM-T2G to specialize in generating graphs aligning with corresponding domains. Moreover, the network-level user prompts in LGGM-T2G$^{UP}$ provide a finer-level control compared to the domain-level descriptions in LGGM-T2G$^D$, further boosting the performance. Furthermore, we shuffle the domain names paired with each graph for LGGM-T2G$^D$ in the testing phase and observe the performance decrease in Table 19 as expected.

LGGM-T2G can also control the properties of the generated graphs. Here, we first synthesize ground-truth graphs with clustering coefficients between [0, 0.75] and average degrees between [0, 100]. We divide these ground-truth graphs into three groups, low/medium/high, and prompt GPT4 to generate user instructions describing these two graph properties (Appendix D.3). Then, we combine these three groups of graphs with their instructions to train LGGM-T2G and evaluate whether the properties of the generated graphs align with the instructions. In Figure 5(a)/(b), we can see a clear alignment between the statistical properties of the ground-truth graphs and generated graphs, both in terms of the average CC and DEG. Moreover, the observed long-tail overlap between the generated graphs in the high and medium CC groups is due to the higher similarity in the embedding space of their conditional text inputs, as illustrated in Figure 5(b). This demonstrates that in the conditional generation, data-quality issues in conditioning input can be reflected in the generated graphs to some extent, highlighting the importance of data-centric approaches in generative tasks (Qin et al., 2023b).

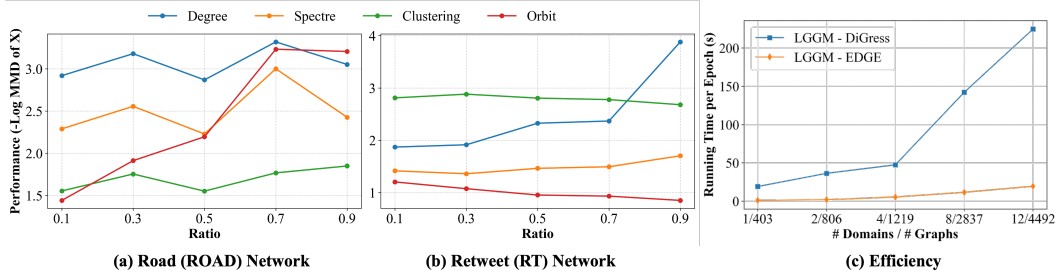

Figure 6: (a) Scaling laws hold in road networks. (b) In retweet networks, scaling laws are observed in Spectre and Degree metrics but not in Clustering and Orbit metrics. (c) Equipping EDGE with our proposed large-scale training paradigm significantly accelerates training compared to DiGress.

## 5.6  ADDITIONAL ANALYSIS

**Data Scaling Law Analysis:** Following the experimental setting for zero-shot generation performance of LGGM in Section 5.2, we increase the size of training graphs for all other domains ranging from ratio 0.1 to ratio 0.9 and show the results for the following two datasets: Road and Retweet in Figure 6. On the road network, we see that the performance gradually increases as the training graphs increase, aligning with the data scaling law. However, on Retweet, although we observe a similar trend in Degree and Spectre metrics, we do not observe a similar trend in Clustering and Orbit.

**Efficiency of LGGM:** We further validate the efficiency of LGGM by visualizing the training time per epoch when equipped with DiGress and EDGE. Results indicate that the primary bottleneck lies in the underlying diffusion model, as evidenced by the significant difference in training time between LGGM-EDGE and LGGM-DiGress. A comprehensive analysis is given in Table 6 in Appendix B.

## 6  CONCLUSION AND FUTURE WORK

Motivated by the recent successes of Large Generative Models across fields of vision, language, video, and audio, and recognizing the promising practical usage of graph generative models, we introduce, for the first time, the large-scale training paradigm that leads to the development of Large Graph Generative Models (LGGMs). These models are trained on over 5,000 graphs sourced from 13 distinct domains from the well-known Network Repository. We empirically verify the superiority of our LGGMs in three aspects. Firstly, our pre-trained LGGM-X models demonstrate exceptional zero-shot generative capabilities. Secondly, LGGMs show remarkable adaptability for fine-tuning, and the fine-tuned LGGM is even more powerful than previous graph generative models trained from scratch. The generated graphs by our LGGMs could boost the training data and lead to better graph classification performance. Lastly, our models facilitate Text-to-Graph generation, enabling users to customize their network generation through prompts.

Looking ahead, we identify several transformative research directions. First, text-controllable graph generation has promising potential for scientific discovery, such as in designing drugs with specific properties (Wang et al., 2022b). Moreover, LGGMs can enhance training data, especially in scenarios with limited graph availability for applications like graph anomaly detection and molecular classification (Liu et al., 2024a; Qin et al., 2023b; Ranshous et al., 2015), as demonstrated by our initial exploration in graph classification (refer to Table 3). Additionally, our Text2-Graph generation reveals that issues present in input conditions can cascade into the quality of generated graphs (long-tail overlap in Figure 5), emphasizing the data-centric perspective in generative artificial intelligence.

## 7  REPRODUCIBILITY

We follow a rigorous reproducibility routine and provide the code and the dataset necessary to reproduce the results shown in Figures 1, 3, 4, and Tables 2, 4 on our GitHub repository: https://github.com/KINDLab-Fly/LGGM. The repository includes the complete framework for pre-training, fine-tuning, and text-to-graph generation. Additionally, we offer a demo at https://lggm-lg.github.io/ that enables users to experiment with our Text2Graph generation, controlling the degree and average clustering coefficient of the generated graphs.

## 8    ACKNOWLEDGEMENT

This research is supported by Adobe Research and the National Science Foundation (NSF) under grant numbers IIS2239881, IIS 2212143 and CAREER Grant No. IIS 1845491. Any opinions, findings, and conclusions or recommendations expressed in this material are those of the author(s) and do not necessarily reflect the views of the National Science Foundation or other funding parties.

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

# Appendix

## Table of Contents

# A  NOTATIONS

This section summarizes the notations used throughout this paper.

Table 5: Notations used throughout this paper.

| Notations | Definitions or Descriptions |
|---|---|
| $\mathbb{G}, \mathbb{G}^c$ | Random variable of universal graphs and graphs from domain $c$ |
| $\mathcal{G}, \mathcal{G}^c$ | Set of universal graphs and graphs from domain c |
| $\mathcal{G}^{\text{Train, Val, Test, c}}$ | Set of training/validation/testing graphs from domain $c$ |
| $P(\mathbb{G}), P(\mathbb{G}^c)$ | Distribution of universal graphs and graphs from domain c |
| $G = (\mathbf{X}^G, \mathbf{E}^G)$ | Graph $G$ with node/edge category matrices $\mathbf{X}^G, \mathbf{E}^G$ |
| $n_G$ | Number of nodes in graph $G$ |
| $d_X/d_E$ | Number of node/edge categories |
| $\mathbf{Q}_X^t, \mathbf{Q}_E^t$ | Node/edge transition matrices |
| $\bar{\mathbf{Q}}_X^t, \bar{\mathbf{Q}}_E^t$ | Node/edge accumulative transition matrices |
| $\mathbf{m}_X^c, \mathbf{m}_E^c$ | Distribution of node/edge categories of graphs from domain c |
| $t, \mathcal{T}$ | Diffusion step $t$ and the set of total steps $\mathcal{T}$ |
| $\widetilde{\mathbb{G}}$ | Distribution of graphs from unseen domains |
| $\widetilde{\mathcal{G}}$ | Set of graphs from unseen domains |
| $S, \phi(S)$ | Text with its embedding from the pre-trained textual encoder $\phi$ |
| $P(\mathbb{G}, \mathbb{S})$ | Joint distribution of graphs and their textual descriptions |
| $\Theta$ | Parameters of Neural Networks |
| $\Theta^\star$ | Optimal Parameters of Neural Networks after pre-training |
| $\Theta^{\star\star}$ | Optimal Parameters of Neural Networks after fine-tuning |
| $\Theta^\square$ | Optimal Parameters of Neural Networks after Text2Graph Generation |
| FB, ASN | Facebook Networks, Animal Social Networks |
| EMAIL, WEB | Email Networks, Web Graphs |
| ROAD, POWER | Road Networks, Power Networks |
| CHEM, BIO | Chemical Networks, Biological Networks |
| ECON, RT | Economic Networks, Retweet Networks |
| COL, ECO | Collaboration Networks, Ecological Networks |
| CITATION | Citation Networks |
| LGGM-X | Pre-trained LGGM on all other domains except X |
| Fine-tuned LGGM on X | Fine-tuned LGGM-X on domain X |
| LGGM-T2G | LGGM trained on graphs paired with texts |
| LGGM-T2G$^D$ | LGGM trained on graphs with texts on domains |
| LGGM-T2G$^{UP}$ | LGGM trained on graphs with user prompts on domains/names |
| LGGM | LGGM trained on all graphs from all domains |

# B  SPACE AND TIME COMPLEXITY ANALYSIS

Table 6: Our theoretical/empirical analysis of the DiGress and EDGE graph diffusion models, both with and without our Large Graph Training Scheme (LGGM). Incorporating LGGM only increases complexity linearly due to the added domains, aligning with the theoretical analysis. $T$ - number of diffusion steps, $\mathcal{V}/\mathcal{E}$ - number of nodes/edges, K - number of active nodes, C - number of domains.

| Backbone | Training Strategy | Theoretical Time Space Complexity | Running Time per Epoch (s) with #Domains/#Graphs | | | | |
|---|---|---|---|---|---|---|---|
| | | | 1/403 | 2/806 | 4/1219 | 8/2837 | 12/4492 |
| DiGress | Original | $\mathcal{O}(T\|\mathcal{V}\|^2)$ | 19.12±0.03 | – | – | – | – |
| | LGGM | $\mathcal{O}(CT\|\mathcal{V}\|^2)$ | 19.14±0.05 | 36.34±0.21 | 47.46±0.11 | 142.14±0.19 | 224.74±0.23 |
| EDGE | Original | $\mathcal{O}(T\max(\|\mathcal{E}\|, K^2))$ | 1.02±0.13 | – | – | – | – |
| | LGGM | $\mathcal{O}(CT\max(\|\mathcal{E}\|, K^2))$ | 1.07±0.18 | 1.92±0.26 | 5.42±0.09 | 11.59±0.20 | 19.48±0.22 |

## C Proof of Theorems

**Theorem 1.** *If the transition matrices* $\mathbf{Q}_X^t, \mathbf{Q}_E^t$ *are independent of the textual description* $\mathbb{S}$*, then we have* $P(\mathbb{G}^{t-1}|\mathbb{G}^t, \mathbb{G}, \mathbb{S}) \propto P(\mathbb{G}^t|\mathbb{G}^{t-1})P(\mathbb{G}^{t-1}|\mathbb{G})$ *and correspondingly, we have the analytical formed solution, i.e.,* $P(\mathbf{X}^{t-1}|\mathbf{X}^t, \mathbf{X}, S) \propto \mathbf{X}^t(\mathbf{Q}_X^t)^\top \odot \mathbf{X}\bar{\mathbf{Q}}_X^{t-1}$, $P(\mathbf{E}^{t-1}|\mathbf{E}^t, \mathbf{E}, S) \propto \mathbf{E}^t(\mathbf{Q}_E^t)^\top \odot$ $\mathbf{E}\bar{\mathbf{Q}}_E^{t-1}$ *following Vignac et al. (2023).*

*Proof.* Applying the Bayes rule, we have:

$$P(\mathbb{G}^{t-1}|\mathbb{G}^t, \mathbb{G}, \mathbb{S}) \propto P(\mathbb{G}^{t-1}, \mathbb{G}^t, \mathbb{G}, \mathbb{S}) \propto P(\mathbb{G}^t|\mathbb{G}^{t-1}, \mathbb{G}, \mathbb{S})P(\mathbb{G}^{t-1}, \mathbb{G}, \mathbb{S}) \tag{6}$$

$$\propto P(\mathbb{G}^t|\mathbb{G}^{t-1}, \mathbb{G}, \mathbb{S})P(\mathbb{G}^{t-1}|\mathbb{G}, \mathbb{S})P(\mathbb{G}, \mathbb{S}). \tag{7}$$

Given the independence of the transition matrix on the textual description $S$ and also the noise is Markovian Vignac et al. (2023), we have $P(\mathbb{G}^t|\mathbb{G}^{t-1}, \mathbb{G}, \mathbb{S}) = P(\mathbb{G}^t|\mathbb{G}^{t-1})$, $P(\mathbb{G}^{t-1}|\mathbb{G}, \mathbb{S}) = P(\mathbb{G}^{t-1}|\mathbb{G})$, and also the irrelevance of $P(\mathbb{G}, \mathbb{S})$ to $P(\mathbb{G}^{t-1}|\mathbb{G}^t, \mathbb{G}, \mathbb{S})$, we then end up with:

$$P(\mathbb{G}^{t-1}|\mathbb{G}^t, \mathbb{G}, \mathbb{S}) \propto P(\mathbb{G}^t|\mathbb{G}^{t-1})P(\mathbb{G}^{t-1}|\mathbb{G}). \tag{8}$$

Since the distribution of graphs can be decomposed into the distribution of node and edge categories, following Vignac et al. (2023), we similarly have:

$$P(\mathbf{X}^{t-1}|\mathbf{X}^t, \mathbf{X}, S) \propto P(\mathbf{X}^t|\mathbf{X}^{t-1})P(\mathbf{X}^{t-1}|\mathbf{X}) = \mathbf{X}^t(\mathbf{Q}_X^t)^\top \odot \mathbf{X}\bar{\mathbf{Q}}_X^{t-1}, \tag{9}$$

$$P(\mathbf{E}^{t-1}|\mathbf{E}^t, \mathbf{E}, S) \propto P(\mathbf{E}^t|\mathbf{E}^{t-1})P(\mathbf{E}^{t-1}|\mathbf{E}) = \mathbf{E}^t(\mathbf{Q}_E^t)^\top \odot \mathbf{E}\bar{\mathbf{Q}}_E^{t-1}. \tag{10}$$

$\square$

**Theorem 2.** *Given the decomposition in Eq. (4) that* $P(\mathbb{G}^{t-1}|\mathbb{G}^t, \mathbb{S}) \propto \sum_{\mathbb{G}} P(\mathbb{G}^{t-1}|\mathbb{G}^t, \mathbb{G}, \mathbb{S})P(\mathbb{G}|\mathbb{G}^t, \mathbb{S})$*, optimizing* $\mathbf{\Theta}$ *according to Eq. (5) essentially optimizes the variational lower bound of the log-likelihood* $P_{\mathbf{\Theta}}(\mathbb{G}^0, \mathbb{S})$*.*

*Proof.* We start directly from the log-likelihood of the joint distribution of $P_{\mathbf{\Theta}}(\mathbb{G}^0, \mathbb{S})$:

$$\log P_{\mathbf{\Theta}}(\mathbb{G}^0, \mathbb{S}) = \log \int P_{\mathbf{\Theta}}(\mathbb{G}^0, \mathbb{S}, \mathbb{G}^1, ..., \mathbb{G}^T)d(\mathbb{G}^1, \mathbb{G}^2, ..., \mathbb{G}^T) \tag{11}$$

$$= \log \int \frac{P_{\mathbf{\Theta}}(\mathbb{G}^0, \mathbb{S}, \mathbb{G}^1, ..., \mathbb{G}^T)}{q(\mathbb{G}^1, \mathbb{G}^2, ..., \mathbb{G}^T)} q(\mathbb{G}^1, \mathbb{G}^2, ..., \mathbb{G}^T)d(\mathbb{G}^1, \mathbb{G}^2, ..., \mathbb{G}^T) \tag{12}$$

$$= \log \mathbb{E}_{q(\mathbb{G}^1, \mathbb{G}^2, ..., \mathbb{G}^T)} \frac{P_{\mathbf{\Theta}}(\mathbb{G}^0, \mathbb{S}, \mathbb{G}^1, ..., \mathbb{G}^T)}{q(\mathbb{G}^1, \mathbb{G}^2, ..., \mathbb{G}^T)} \tag{13}$$

$$\geq \mathbb{E}_{q(\mathbb{G}^1, \mathbb{G}^2, ..., \mathbb{G}^T)} \log \frac{P_{\mathbf{\Theta}}(\mathbb{G}^0, \mathbb{S}, \mathbb{G}^1, ..., \mathbb{G}^T)}{q(\mathbb{G}^1, \mathbb{G}^2, ..., \mathbb{G}^T)} \quad \text{by Jensen's inequality} \tag{14}$$

$$= \mathbb{E}_{q(\mathbb{G}^1, \mathbb{G}^1, ..., \mathbb{G}^T)} \log \frac{P(\mathbb{G}^T, \mathbb{S}) \prod_{t=1}^T P_{\mathbf{\Theta}}(\mathbb{G}^{t-1}|\mathbb{G}^t, \mathbb{S})}{q(\mathbb{G}^1) \prod_{t=2}^T q(\mathbb{G}^t|\mathbb{G}^{t-1})} \quad \text{by Markovian} \tag{15}$$

$$= \mathbb{E}_{q(\mathbb{G}^0, \mathbb{G}^1, ..., \mathbb{G}^T)}[\log P(\mathbb{G}^T, \mathbb{S}) + \sum_{t=1}^T \log \frac{P_{\mathbf{\Theta}}(\mathbb{G}^{t-1}|\mathbb{G}^t, \mathbb{S})}{q(\mathbb{G}^t|\mathbb{G}^{t-1})}] + \text{const.} \tag{16}$$

According to the decomposition in Eq. (2), optimizing $\mathbf{\Theta}$ according to Eq. (5) leads to optimizing $P_{\mathbf{\Theta}}(\mathbb{G}^{t-1}|\mathbb{G}^t, \mathbb{S})$, which corresponds to the second term in Eq. (16) and subsequently optimizes the variational lower bound of the log-likelihood $P_{\mathbf{\Theta}}(\mathbb{G}^0, \mathbb{S})$ according to the derivation from Eq. (11) to Eq. (16). Therefore, training Text-to-Graph LGGM according to Eq. (5) enables the model to generate graphs such that the pairs of texts and graphs end up with higher likelihoods.

$\square$

# D  DATA PREPARATION

## D.1  PRE-PROCESSED GRAPHS FOR TRAINING LGGMS

We select graphs from the Network Repository across 13 distinct yet representative domains covering a wide variety of real-world scenarios, including Facebook, Animal Social, Email, Web, Road, Power, Chemical, Biological, Economic, Retweet, Collaboration, Ecological, and Citation. Due to the scalability with diffusion-based graph generative models, we further sample subgraphs for certain domains, and Table 7 presents the comprehensive statistics of the sampled subgraphs, which are used for training LGGMs. We can see that graphs from different domains are statistically different.

Table 7: Summary of Graph Statistics. Facebook (FB), Animal Social (ASN), Email, Web, Road, Power, Chemical (CHEM), Biological (BIO), Economic (ECON), Retweet (RT), Collaboration (COL), Ecological (ECO), Citation.

| Category | Num Nodes | Num Edges | Avg Degree | Avg Clustering | Max Nodes | Min Nodes | Max Edges | Min Edges | Num Graphs |
|---|---|---|---|---|---|---|---|---|---|
| ASN | $52.47 \pm 40.13$ | $77.59 \pm 80.95$ | $2.62 \pm 1.52$ | $0.395 \pm 0.178$ | 283 | 3 | 515 | 2 | 267 |
| BIO | $191.14 \pm 43.47$ | $965.71 \pm 878.35$ | $9.16 \pm 7.69$ | $0.276 \pm 0.199$ | 258 | 109 | 4392 | 96 | 504 |
| CHEM | $36.46 \pm 20.49$ | $64.61 \pm 26.23$ | $3.75 \pm 0.63$ | $0.421 \pm 0.223$ | 125 | 2 | 149 | 1 | 646 |
| Citation | $235.91 \pm 27.25$ | $1287.16 \pm 1087.00$ | $10.17 \pm 8.14$ | $0.369 \pm 0.224$ | 270 | 175 | 4474 | 188 | 504 |
| COL | $174.26 \pm 53.82$ | $312.56 \pm 176.33$ | $3.41 \pm 1.24$ | $0.497 \pm 0.203$ | 247 | 52 | 996 | 68 | 504 |
| ECO | $100.67 \pm 30.10$ | $1490.00 \pm 673.87$ | $27.72 \pm 7.00$ | $0.406 \pm 0.082$ | 128 | 54 | 2106 | 353 | 6 |
| ECON | $144.18 \pm 35.82$ | $3258.76 \pm 3540.28$ | $39.76 \pm 37.80$ | $0.419 \pm 0.296$ | 219 | 90 | 11142 | 188 | 504 |
| Email | $146.67 \pm 35.86$ | $681.55 \pm 500.28$ | $9.79 \pm 7.26$ | $0.389 \pm 0.211$ | 213 | 82 | 2909 | 216 | 504 |
| Power | $132.22 \pm 20.29$ | $289.32 \pm 183.02$ | $4.35 \pm 2.31$ | $0.161 \pm 0.164$ | 187 | 81 | 1332 | 133 | 512 |
| Road | $265.25 \pm 94.31$ | $276.46 \pm 79.61$ | $2.70 \pm 2.08$ | $0.078 \pm 0.134$ | 411 | 32 | 456 | 137 | 504 |
| RT | $104.11 \pm 35.23$ | $110.99 \pm 46.44$ | $2.11 \pm 0.37$ | $0.028 \pm 0.038$ | 175 | 35 | 295 | 34 | 558 |
| FB | $219.45 \pm 47.05$ | $1863.44 \pm 701.53$ | $16.36 \pm 6.17$ | $0.315 \pm 0.083$ | 259 | 48 | 3898 | 46 | 504 |
| Web | $173.32 \pm 24.86$ | $462.21 \pm 336.46$ | $5.09 \pm 3.06$ | $0.404 \pm 0.196$ | 231 | 119 | 1607 | 149 | 504 |

## D.2  PREPARING GRAPHS AND TEXT DESCRIPTION ABOUT THEIR DOMAINS/NAMES

Here we thoroughly discuss the process of obtaining graphs and their corresponding text prompts describing their domains/names. As given by the Network Repository, we directly download graphs along with their domains/names. We then prompt GPT3.5 to generate user prompts describing the graph given its domain/name. The concrete prompt template we use here is shown in Listing 1 with exemplary generated user prompts shown in Listing 2. Moreover, we apply the sentence transformer to obtain text embeddings of the generated prompts for each network and perform t-SNE visualization. As shown in Figure 7a, we see prompts for graphs from different domains from different clusters. More importantly, textual similarity can somewhat reflect their network similarity. For example, prompts for road and power networks are very close, and they both belong to infrastructure. Moreover, Facebook Networks, Email Networks, Collaboration Networks, Web Graphs are very close since all these four belong to some sub-variants of social networks. *This inherent relationship between the textual similarity and structural similarity between two graphs demonstrates that the world knowledge encoded in the text could somehow provide useful preference for the graphs to be generated.*

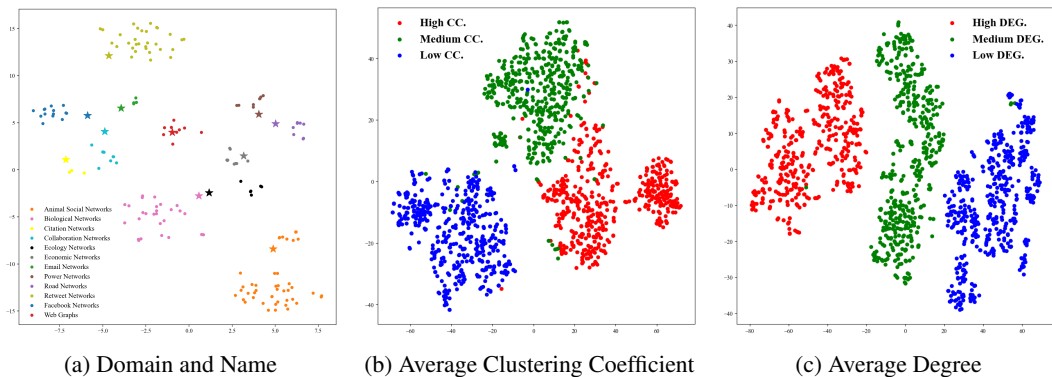

(a) Domain and Name  (b) Average Clustering Coefficient  (c) Average Degree

Figure 7: t-SNE visualization of textual description about network (a) domain/name (b) average clustering coefficient (c) average degree.

Listing 1: Prompt Template for Generating Textual Description about Network Domain/Name.

PROMPT: Given a graph called {GRAPH NAME} that is from the DOMAIN {DOMAIN NAME}. Please
    generate a short sentence about the graph.

Note:
∗ Do not generate more than 20 words.

Listing 2: Examples of Textual Description about Network Domain/Name.

```
===============================================================================
∗DOMAIN: Animal Social Networks
∗NAME: reptilia−tortoise−network−sl
∗TEXT: The reptilia−tortoise−network−sl graph represents the social connections among tortoises in the
    reptile community.
===============================================================================
∗DOMAIN: Power Networks
∗NAME: power−eris1176
∗TEXT: The power−eris1176 graph represents the interconnected nodes and edges of a power network system
===============================================================================
∗DOMAIN: Economic Networks
∗NAME: econ−poli
∗TEXT: The econ−poli graph represents the interconnectedness of economic and political factors in a
    network.
===============================================================================
∗DOMAIN: Ecology Networks
∗NAME: eco−evergla
∗TEXT: The eco−evergla graph represents the interconnectedness of species in the Everglades ecosystem.
===============================================================================
∗DOMAIN: Email Networks
∗NAME: email−enron−only
∗TEXT: The email−enron−only graph represents the network of email communication within the Enron
    corporation.
===============================================================================
∗DOMAIN: Road Networks
∗NAME: road−roadNet−CA
∗TEXT: The road−roadNet−CA graph represents the road network in California.
===============================================================================
∗DOMAIN: Retweet Networks
∗NAME: rt_occupywallstnyc
∗TEXT: The graph rt_occupywallstnyc represents retweet relationships in the Occupy Wall Street movement
    in New York City.
===============================================================================
∗DOMAIN: Facebook Networks
∗NAME: socfb−Haverford76
∗TEXT: The socfb−Haverford76 graph represents the social connections among users in the Haverford
    College community on Facebook.
===============================================================================
∗DOMAIN: Web Graphs
∗NAME: web−wiki−chameleon
∗TEXT: The web−wiki−chameleon graph represents the interconnections between web pages, Wikipedia
    articles, and chameleon species.
===============================================================================
∗DOMAIN: Biological Networks
∗NAME: bio−WormNet−v3−benchmark
∗TEXT: The bio−WormNet−v3−benchmark graph represents a biological network related to worms.
===============================================================================
∗DOMAIN: Citation Networks
∗NAME: cit−DBLP
∗TEXT: cit−DBLP is a graph representing the citation relationships between research papers in the field of
    computer science.
===============================================================================
∗DOMAIN: Collaboration Networks
∗NAME: ca−netscienc
∗TEXT: The ca−netscienc graph represents a collaboration network in the field of science.
===============================================================================
```

### D.3 PREPARING GRAPHS AND THEIR TEXTUAL DESCRIPTION ABOUT GRAPH PROPERTY

Here we thoroughly discuss the process of obtaining graphs and their corresponding text prompts describing their properties. Our goal is to demonstrate that Text2Graph LGGM can control the statistics of the generated graphs in the full spectrum. However, the graphs obtained directly from the Network Repository do not cover the whole topological space (e.g., Figure 1(a) shows that no networks have a higher average degree while low clustering coefficient). Therefore, we plan to synthesize graphs covering the whole space by Watts-Strogatz Small-world Graph Model. We vary the number of nodes between [10, 110], the number of initial neighbors between [5, number of nodes], and also the probability of rewiring each edge between [0, 1] to ensure the generated graphs span across the full spectrum. After that, we group the generated graphs into low, medium, and high groups in terms of their clustering coefficient and average degree. We implement this using NetworkX.

After we synthesize graphs and divide them into three groups, we generate user prompts paired with these graphs next. Specifically, we prompt GPT4 following the templates in Listing 3/4. To ensure the compatibility between the synthesis graphs and the generated user prompts. We further replace the number output by GPT4 describing the network property with the real statistic calculated from each network.

Listing 3: Prompt Template for Generating Textual Description about Network Property.

```
================================================================================
PROMPT: Please generate a short sentence about the graph, including its clustering coefficient information.

Note:
∗ Do not generate more than 20 words.
∗ Make sure the generated sentence includes the level of clustering coefficient, you can either specify it via
      words like ['low', 'medium', 'high']. or specify it via numbers like [(0, 0.25), (0.25, 0.5), (0.5, 0.75)]"
∗ You can also sometimes specify a concrete application scenario of the generated network.
∗ Please be accurate but also diverse
================================================================================
PROMPT: Please generate a short sentence about the graph, including its average degree information.

Note:
∗ Do not generate more than 20 words.
∗ Make sure the generated sentence includes the level of average degree, you can either specify it via words
      like ['low', 'medium', 'high']. or specify it via numbers like [(0, 20), (20, 50), (50, 100)]"
∗ You can also sometimes specify a concrete application scenario of the generated network.
∗ Please be accurate but also diverse
================================================================================
```

Listing 4: Examples of Textual Description about Network Property.

```
================================================================================
∗ This graph has a high clustering coefficient, suggesting strong node clustering.
∗ Please generate a network with a clustering coefficient around 0.61, indicating strong clustering.
∗ This retirement community's social interaction graph displays a high clustering coefficient of 0.73,
      indicative of close relationships.
================================================================================
∗ With an average degree of 35, this network is ideal for studying urban transportation patterns.
∗ The graph's moderate connectivity level helps in understanding the structure of small to medium−sized
      music bands.
∗ An average degree of 41 makes this network suitable for simulating the collaboration in local artisan
      markets.
================================================================================
```

# E    EXPERIMENTAL SETTING

## E.1    EVALUATION METRICS

Following Thompson et al. (2022); You et al. (2018), we evaluate the graph generation performance by the standard Maximum Mean Discrepancy (MMD) between generated and reference graphs $\mathcal{G}_g, \mathcal{G}_r$:

$$\mathrm{MMD}(\mathcal{G}_g, \mathcal{G}_r) = \frac{1}{m^2} \sum_{i,j=1}^{m} k(\mathbf{x}_i^r, \mathbf{x}_j^r) + \frac{1}{n^2} \sum_{i,j=1}^{n} k(\mathbf{x}_i^g, \mathbf{x}_j^g) - \frac{2}{nm} \sum_{i=1}^{n} \sum_{j=1}^{m} k(\mathbf{x}_i^g, \mathbf{x}_j^r), \quad (17)$$

where $k(\cdot, \cdot)$ is a general kernel function and specifically we use RBF kernel following You et al. (2018):

$$k(\mathbf{x}_i, \mathbf{x}_j) = \exp(-d(\mathbf{x}_i, \mathbf{x}_j)/2\sigma^2), \quad (18)$$

where $d(\cdot, \cdot)$ computes pairwise distance following Vignac et al. (2023) and MMD is evaluated over the distributions of degree (DEG), clustering coefficients (CC), eigenvalues of normalized Laplacian matrix (Spec) and orbits counts representing the distribution of all substructures of size 4 (Orb).

## E.2    HYPERPARAMETER DETAILS

For all experiments, we select the best configuration according to the generation performance on validation graphs and report the final performance on generating testing graphs. We adopt the default hyperparameter settings from DiGress Vignac et al. (2023) with the following exceptions: we generate 100 graphs per domain for each evaluation and set the training epochs at 300 to ensure convergence. Additionally, we implement gradient accumulation, using a mini-batch size of 12 across 4 accumulations, resulting in an effective batch size of 48. For Text-to-Graph Generation, the textual encoder used to obtain textual description embeddings is "all-MiniLM-L6-v2". All experiments are performed on a machine with A100-80G GPU RAM and 128GB RAM.

## E.3    PARADIGM SETUP

Figure 8 comprehensively visualizes the training/evaluation paradigms of the four experiments, the details of which are discussed in Section 5.1.

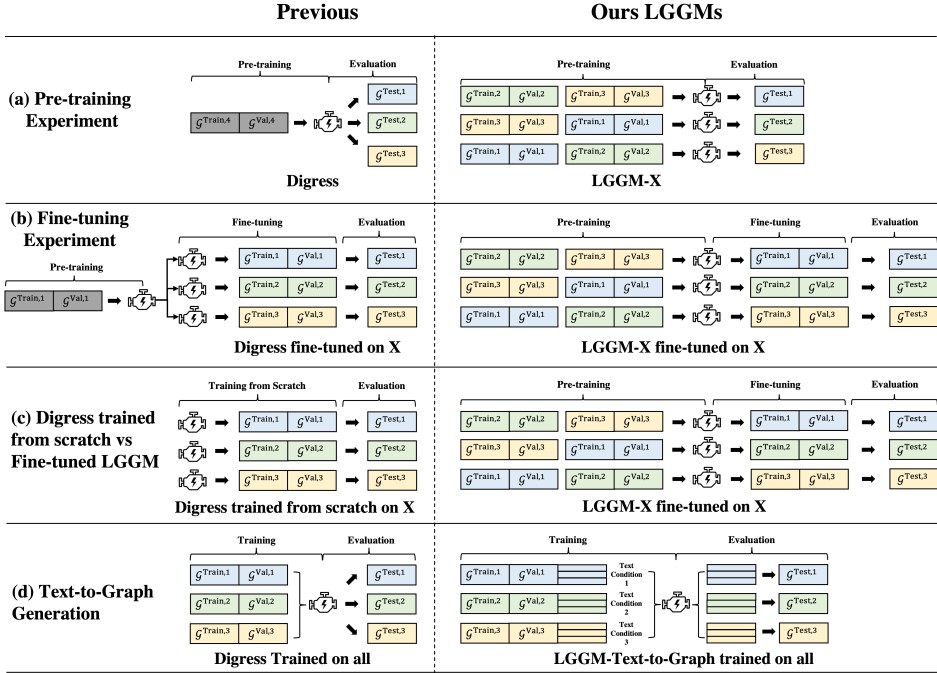

Figure 8: Comprehensive Overview of the Experimental Setup for our LGGMs.

# F FULL EXPERIMENTAL RESULTS

## F.1 OUT-OF-DOMAIN PERFORMANCE COMPARISON BETWEEN DIGRESS AND LGGM

### F.1.1 DOMAIN SPECIFIC TRANSITION STRATEGY

Table 8: Comparing Zero-shot Generation Performance on Unseen Graphs in domain X between DiGress trained on QM9 and LGGM-X pre-trained on all domains except the held-out domain X.

| Domain | Method | DEG | CC | Spec | Orb | Domain | Method | DEG | CC | Spec | Orb |
|---|---|---|---|---|---|---|---|---|---|---|---|
| FB | DiGress | **0.2695** | **0.3452** | **0.0649** | **0.1489** | BIO | DiGress | 0.2419 | **0.2993** | **0.1101** | **0.2978** |
| | LGGM-X | 0.4962 | 0.7625 | 0.3408 | 0.7982 | | LGGM-X | **0.2117** | 0.6365 | 0.1690 | 0.5156 |
| ASN | DiGress | 0.1793 | 0.4721 | 0.1751 | 0.5654 | ECON | DiGress | 0.2811 | 0.2042 | 0.2028 | 0.2633 |
| | LGGM-X | **0.0220** | **0.4044** | **0.1274** | **0.0505** | | LGGM-X | **0.1916** | **0.0917** | **0.1219** | **0.0640** |
| EMAIL | DiGress | **0.2312** | **0.5444** | **0.0674** | **0.2650** | RT | DiGress | 0.4466 | 0.4170 | 0.4483 | 0.4551 |
| | LGGM-X | 0.2618 | 0.8650 | 0.3013 | 1.0459 | | LGGM-X | **0.0721** | **0.0517** | **0.2331** | **0.4085** |
| WEB | DiGress | 0.2575 | **0.5955** | 0.1907 | 0.9282 | COL | DiGress | 0.2393 | **0.5341** | 0.2247 | 0.7619 |
| | LGGM-X | **0.1491** | 0.9436 | **0.1154** | **0.4016** | | LGGM-X | **0.1493** | 0.9200 | **0.1786** | **0.2057** |
| ROAD | DiGress | 0.4111 | 0.6653 | 0.3084 | 0.6530 | ECO | DiGress | 0.4580 | 0.4546 | 0.2144 | 0.4417 |
| | LGGM-X | **0.0379** | **0.1191** | **0.0759** | **0.0401** | | LGGM-X | **0.2049** | **0.2760** | **0.0691** | **0.2107** |
| POWER | DiGress | 0.5292 | **0.6083** | 0.3556 | 1.2124 | CITATION | DiGress | 0.3159 | **0.3664** | 0.1299 | **0.2278** |
| | LGGM-X | **0.0343** | 0.6290 | **0.0649** | **0.0228** | | LGGM-X | **0.1314** | 0.8908 | **0.1188** | 0.6391 |
| ALL | DiGress | 0.3217 | **0.4589** | 0.2077 | 0.5184 | | | | | | |
| | LGGM-X | **0.1635** | 0.5492 | **0.1597** | **0.3669** | | | | | | |

### F.1.2 UNIFORM TRANSITION STRATEGY

Table 9: Comparing Zero-shot Generation Performance on Unseen Graphs in domain X between DiGress trained on QM9 and LGGM-X pre-trained on all domains except the held-out domain X.

| Domain | Method | DEG | CC | Spec | Orb | Domain | Method | DEG | CC | Spec | Orb |
|---|---|---|---|---|---|---|---|---|---|---|---|
| FB | DiGress | **0.3376** | **0.6298** | **0.0797** | **0.3593** | BIO | DiGress | 0.2712 | 0.5202 | 0.1127 | 0.3188 |
| | LGGM-X | 0.4723 | 0.6843 | 0.2924 | 0.7555 | | LGGM-X | **0.1081** | **0.2696** | **0.0900** | **0.2053** |
| ASN | DiGress | 0.1496 | 0.3258 | 0.1506 | 0.4420 | ECON | DiGress | 0.2987 | 0.4841 | 0.2162 | 0.3834 |
| | LGGM-X | **0.0281** | **0.2440** | **0.0830** | **0.0618** | | LGGM-X | **0.1213** | **0.0920** | **0.1120** | **0.1086** |
| EMAIL | DiGress | 0.2192 | 0.6012 | 0.0702 | 0.3416 | RT | DiGress | 0.4164 | 0.1327 | 0.4147 | 0.5957 |
| | LGGM-X | **0.0751** | **0.2364** | **0.0768** | **0.3089** | | LGGM-X | **0.0525** | **0.1429** | **0.1330** | **0.2219** |
| WEB | DiGress | 0.2556 | 0.6186 | 0.1877 | 0.6045 | COL | DiGress | 0.2473 | 0.5826 | 0.2314 | 0.7679 |
| | LGGM-X | **0.0648** | **0.3961** | **0.0549** | **0.1127** | | LGGM-X | **0.0736** | **0.5769** | **0.0895** | **0.0988** |
| ROAD | DiGress | 0.3705 | 0.8226 | 0.2801 | 0.7198 | ECO | DiGress | 0.5431 | 0.7915 | 0.2338 | 0.6045 |
| | LGGM-X | **0.0713** | **0.2193** | **0.0987** | **0.2986** | | LGGM-X | **0.4753** | **0.3904** | 0.3194 | **0.3934** |
| POWER | DiGress | 0.3726 | 0.4582 | 0.3270 | 1.4732 | CITATION | DiGress | 0.2527 | 0.7790 | 0.1315 | **0.4966** |
| | LGGM-X | **0.0119** | **0.1293** | **0.0373** | **0.0754** | | LGGM-X | **0.1348** | **0.7257** | **0.1160** | 0.4981 |
| ALL | DiGress | 0.3112 | 0.5622 | 0.2030 | 0.5923 | | | | | | |
| | LGGM-X | **0.1408** | **0.3422** | **0.1253** | **0.2616** | | | | | | |

## F.2 Performance Comparison between Fine-tuned DiGress and Fine-tuned LGGM

### F.2.1 Domain Specific Transition Strategy

Table 10: Comparing Graph Generation Performance between Fine-tuned DiGress and Fine-tuned LGGM on each domain. DiGress-FT: DiGress pre-trained on QM9 and fine-tuned on domain X; LGGM-FT: LGGM pre-trained on all other domains except X and fine-tuned on X under **Domain Specific Transition Strategy**.

| Domain | Method | DEG | CC | Spec | Orb | Domain | Method | DEG | CC | Spec | Orb |
|---|---|---|---|---|---|---|---|---|---|---|---|
| FB | DiGress-FT | 0.0159 | 0.0564 | 0.0082 | 0.0298 | BIO | DiGress-FT | 0.0391 | 0.0354 | 0.0347 | 0.0291 |
| | LGGM-FT | **0.0065** | **0.0544** | **0.0069** | **0.0282** | | LGGM-FT | **0.0036** | **0.0303** | **0.0102** | **0.0342** |
| ASN | DiGress-FT | 0.0189 | 0.0775 | 0.0729 | 0.0886 | ECON | DiGress-FT | 0.0301 | 0.0431 | 0.0372 | 0.0392 |
| | LGGM-FT | **0.0014** | **0.0509** | **0.0161** | **0.0084** | | LGGM-FT | **0.0215** | **0.0330** | **0.0062** | **0.0249** |
| EMAIL | DiGress-FT | 0.0208 | 0.0448 | 0.0230 | **0.0447** | RT | DiGress-FT | 0.0054 | 0.0464 | 0.0051 | 0.0437 |
| | LGGM-FT | **0.0166** | **0.0364** | **0.0104** | 0.0463 | | LGGM-FT | **0.0012** | **0.0075** | **0.0033** | **0.0162** |
| WEB | DiGress-FT | 0.0192 | 0.0808 | 0.0664 | 0.1361 | COL | DiGress-FT | 0.0255 | 0.2279 | 0.0788 | 0.0731 |
| | LGGM-FT | **0.0116** | **0.0721** | **0.0152** | **0.0656** | | LGGM-FT | **0.0202** | **0.1621** | **0.0571** | **0.0631** |
| ROAD | DiGress-FT | 0.0907 | 0.1404 | 0.1099 | 0.1097 | ECO | DiGress-FT | 0.1370 | 0.2747 | 0.0476 | 0.2109 |
| | LGGM-FT | **0.0088** | **0.1349** | **0.0347** | **0.0125** | | LGGM-FT | **0.0196** | **0.2343** | **0.0291** | **0.2100** |
| POWER | DiGress-FT | 0.0104 | 0.2197 | 0.1023 | 0.0445 | CITATION | DiGress-FT | 0.0363 | 0.1140 | 0.0469 | 0.0423 |
| | LGGM-FT | **0.0008** | **0.1539** | **0.0215** | **0.0081** | | LGGM-FT | **0.0078** | **0.0827** | **0.0137** | **0.0316** |
| All | DiGress-FT | 0.0374 | 0.1134 | 0.0528 | 0.0743 | | | | | | |
| | LGGM-FT | **0.0010** | **0.0877** | **0.0187** | **0.0458** | | | | | | |

### F.2.2 Uniform Transition Strategy

Table 11: Comparing Graph Generation Performance between Fine-tuned DiGress and Fine-tuned LGGM on each domain. DiGress-FT: DiGress pre-trained on QM9 and fine-tuned on domain X; LGGM-FT: LGGM pre-trained on all other domains except X and fine-tuned on X under **Uniform Transition Strategy**.

| Domain | Method | DEG | CC | Spec | Orb | Domain | Method | DEG | CC | Spec | Orb |
|---|---|---|---|---|---|---|---|---|---|---|---|
| FB | DiGress-FT | **0.0039** | 0.0650 | 0.0090 | 0.0304 | BIO | DiGress-FT | 0.0274 | 0.0845 | 0.0493 | 0.0312 |
| | LGGM-FT | 0.0050 | **0.0579** | **0.0059** | **0.0280** | | LGGM-FT | **0.0049** | **0.0496** | **0.0056** | **0.0257** |
| ASN | DiGress-FT | 0.0249 | 0.5604 | 0.0779 | 0.0348 | ECON | DiGress-FT | **0.0133** | **0.0355** | 0.0223 | **0.0360** |
| | LGGM-FT | **0.0058** | **0.1098** | **0.0311** | **0.0101** | | LGGM-FT | 0.0597 | 0.0594 | **0.0216** | 0.0535 |
| EMAIL | DiGress-FT | 0.0134 | 0.0709 | 0.0223 | 0.0694 | RT | DiGress-FT | 0.0418 | 0.0243 | 0.0495 | 0.0583 |
| | LGGM-FT | **0.0120** | **0.0559** | **0.0158** | **0.0444** | | LGGM-FT | **0.0032** | **0.0163** | **0.0051** | **0.0227** |
| WEB | DiGress-FT | 0.0327 | 0.2025 | 0.0858 | 0.2033 | COL | DiGress-FT | **0.0562** | 0.7070 | **0.1086** | 0.1471 |
| | LGGM-FT | **0.0218** | **0.1398** | **0.0310** | **0.1262** | | LGGM-FT | 0.1074 | **0.4265** | 0.1398 | **0.0897** |
| ROAD | DiGress-FT | 0.0843 | 0.1010 | 0.1873 | 0.5155 | ECO | DiGress-FT | 0.1118 | 0.3016 | 0.0548 | 0.2102 |
| | LGGM-FT | **0.0081** | **0.0547** | **0.0573** | **0.0228** | | LGGM-FT | **0.0204** | **0.2347** | **0.0404** | **0.2100** |
| POWER | DiGress-FT | 0.0231 | 0.1029 | 0.0683 | 0.0441 | CITATION | DiGress-FT | 0.0277 | 0.1622 | 0.0501 | 0.0813 |
| | LGGM-FT | **0.0077** | **0.0570** | **0.0134** | **0.0040** | | LGGM-FT | **0.0052** | **0.0821** | **0.0221** | **0.0443** |
| All | DiGress-FT | 0.0384 | 0.2015 | 0.0654 | 0.1218 | | | | | | |
| | LGGM-FT | **0.0218** | **0.1120** | **0.0324** | **0.0568** | | | | | | |

### F.3 PERFORMANCE COMPARISON BETWEEN DIGRESS DIRECTLY TRAINED ON X AND FINE-TUNED LGGM

#### F.3.1 DOMAIN SPECIFIC TRANSITION

Table 12: Comparing Graph Generation Performance between DiGress and Fine-tuned LGGM on each domain. DiGress: DiGress trained directly on domain X; LGGM-FT: LGGM pre-trained on all other domains except X and fine-tuned on X under **Domain Specific Transition Strategy**.

| Domain | Method | DEG | CC | Spec | Orb | Domain | Method | DEG | CC | Spec | Orb |
|---|---|---|---|---|---|---|---|---|---|---|---|
| FB | DiGress | 0.0423 | 0.0718 | 0.0243 | 0.0298 | BIO | DiGress | 0.0481 | 0.1286 | 0.0487 | 0.0460 |
| | LGGM-FT | **0.0065** | **0.0544** | **0.0069** | **0.0282** | | LGGM-FT | **0.0036** | **0.0303** | **0.0102** | **0.0342** |
| ASN | DiGress | 0.0319 | 0.0835 | 0.0679 | 0.1463 | ECON | DiGress | 0.0224 | 0.0361 | 0.0084 | 0.0325 |
| | LGGM-FT | **0.0014** | **0.0509** | **0.0161** | **0.0084** | | LGGM-FT | **0.0215** | **0.0330** | **0.0062** | **0.0249** |
| EMAIL | DiGress | **0.0145** | 0.0671 | 0.0143 | 0.0558 | RT | DiGress | 0.0035 | 0.0111 | 0.0094 | 0.0207 |
| | LGGM-FT | 0.0166 | **0.0364** | **0.0104** | **0.0463** | | LGGM-FT | **0.0012** | **0.0075** | **0.0033** | **0.0162** |
| WEB | DiGress | 0.0204 | 0.0778 | 0.0695 | 0.1101 | COL | DiGress | 0.0278 | 0.2192 | 0.0669 | 0.0284 |
| | LGGM-FT | **0.0116** | **0.0721** | **0.0152** | **0.0656** | | LGGM-FT | **0.0202** | **0.1621** | **0.0571** | **0.0631** |
| ROAD | DiGress | 0.0333 | **0.1342** | 0.0932 | 0.0861 | ECO | DiGress | 0.0268 | 0.2356 | 0.0339 | 0.2100 |
| | LGGM-FT | **0.0088** | 0.1349 | **0.0347** | **0.0125** | | LGGM-FT | **0.0196** | **0.2343** | **0.0291** | 0.2100 |
| POWER | DiGress | 0.0143 | 0.2050 | 0.0776 | 0.0392 | CITATION | DiGress | 0.0406 | 0.1790 | 0.0677 | 0.0944 |
| | LGGM-FT | **0.0008** | **0.1539** | **0.0215** | **0.0081** | | LGGM-FT | **0.0078** | **0.0827** | **0.0137** | **0.0316** |
| All | DiGress | 0.0272 | 0.1208 | 0.0485 | 0.0749 | | | | | | |
| | LGGM-FT | **0.0100** | **0.0877** | **0.0187** | **0.0458** | | | | | | |

#### F.3.2 UNIFORM TRANSITION

Table 13: Comparing Graph Generation Performance between DiGress and Fine-tuned LGGM on each domain. DiGress: DiGress trained directly on domain X; LGGM-FT: LGGM pre-trained on all other domains except X and fine-tuned on X under **Uniform Transition Strategy**.

| Domain | Method | DEG | CC | Spec | Orb | Domain | Method | DEG | CC | Spec | Orb |
|---|---|---|---|---|---|---|---|---|---|---|---|
| FB | DiGress | 0.0177 | 0.0698 | 0.0138 | 0.0296 | BIO | DiGress | 0.0179 | 0.0499 | 0.0441 | 0.0526 |
| | LGGM-FT | **0.0050** | **0.0579** | **0.0059** | **0.0280** | | LGGM-FT | **0.0049** | **0.0496** | **0.0056** | **0.0257** |
| ASN | DiGress | 0.0337 | 0.1744 | 0.0482 | 0.0243 | ECON | DiGress | **0.0229** | **0.0430** | **0.0088** | **0.0427** |
| | LGGM-FT | **0.0058** | **0.1098** | **0.0311** | **0.0101** | | LGGM-FT | 0.0597 | 0.0594 | 0.0216 | 0.0535 |
| EMAIL | DiGress | 0.0259 | 0.0901 | 0.0366 | 0.0743 | RT | DiGress | 0.0336 | 0.0920 | 0.0432 | 0.0572 |
| | LGGM-FT | **0.0120** | **0.0559** | **0.0158** | **0.0444** | | LGGM-FT | **0.0032** | **0.0163** | **0.0051** | **0.0227** |
| WEB | DiGress | 0.0239 | **0.0898** | 0.1033 | 0.2371 | COL | DiGress | **0.0252** | 0.5156 | **0.1171** | 0.2060 |
| | LGGM-FT | **0.0218** | 0.1398 | **0.0310** | **0.1262** | | LGGM-FT | 0.1074 | **0.4265** | 0.1398 | **0.0897** |
| ROAD | DiGress | 0.1553 | 0.2788 | 0.2169 | 0.0542 | ECO | DiGress | 0.0263 | 0.2359 | 0.0439 | 0.2100 |
| | LGGM-FT | **0.0081** | **0.0547** | **0.0573** | **0.0228** | | LGGM-FT | **0.0204** | **0.2347** | **0.0404** | 0.2100 |
| POWER | DiGress | 0.0348 | 0.3174 | 0.1083 | 0.1393 | CITATION | DiGress | 0.0217 | 0.1566 | 0.0645 | 0.1235 |
| | LGGM-FT | **0.0077** | **0.0570** | **0.0134** | **0.0040** | | LGGM-FT | **0.0052** | **0.0821** | **0.0221** | **0.0443** |
| All | DiGress | 0.0366 | 0.1761 | 0.0707 | 0.1042 | | | | | | |
| | LGGM-FT | **0.0218** | **0.1120** | **0.0324** | **0.0568** | | | | | | |

## F.4 DOMAIN TRANSFERABILITY ANALYSIS

Table 14: Transferability analysis between Chemistry (CHEM) and Society (SOC) domains. The pre-trained LGGM on chemistry demonstrates negative transferability on IMDB-BINARY/MULTI graphs in the SOC domain. LGGM pre-trained on society demonstrates negative transferability on graphs PROTEINS/ENZYMES/MUTAG in CHEM domain.

| Domain | Chemistry | | | | | | Social | | | |
|--------|-----------|----|------|----|------|----|--------|----|------|----|
| Dataset | PROTEINS | | ENZYMES | | MUTAG | | IMDB-BINARY | | IMDB-MULTI | |
| Metric | Orb | CC | Orb | CC | Orb | CC | Orb | CC | Orb | CC |
| CHEM | 0.0604 | 0.0297 | 0.0593 | 0.0534 | 0.0445 | 0.0340 | 0.9001 | 0.4085 | 0.5511 | 0.6324 |
| SOC | 0.6997 | 0.0890 | 0.8028 | 0.0422 | 0.5022 | 0.9439 | 0.1526 | 0.2247 | 0.0605 | 0.0945 |

## F.5 EQUIPPING LARGE-SCALE TRAINING PARADIGM WITH ANOTHER GRAPH GENERATIVE BACKBONE EDGE

Table 15: Comparing Graph Generation Performance between EDGE and EDGE equipped with LGGM on each domain. We can still see the performance boost after equipping EDGE with our large-scale training paradigm.

| Domain | Method | DEG | CC | Spec | Orb | Domain | Method | DEG | CC | Spec | Orb |
|--------|--------|-----|----|------|-----|--------|--------|-----|----|------|-----|
| FB | EDGE | 0.0031 | **0.0609** | 0.0079 | 0.0362 | BIO | EDGE | 0.0126 | **0.0555** | 0.0484 | 0.0612 |
| | **LGGM** | **0.0022** | 0.0657 | **0.0073** | **0.0354** | | **LGGM** | **0.0120** | 0.0669 | 0.0502 | **0.0590** |
| ASN | EDGE | 0.0212 | 0.1416 | 0.1145 | 0.1652 | ECON | EDGE | **0.0416** | **0.0398** | **0.0078** | **0.0364** |
| | **LGGM** | **0.0146** | **0.0783** | **0.0724** | **0.1285** | | LGGM | 0.0519 | 0.0817 | 0.0665 | 0.0551 |
| EMAIL | EDGE | 0.0118 | 0.0661 | 0.0249 | 0.0771 | RT | EDGE | 0.0340 | **0.1760** | 0.1242 | **0.0331** |
| | **LGGM** | **0.0081** | **0.0519** | **0.0237** | **0.0691** | | **LGGM** | **0.0288** | 0.3088 | **0.0366** | 0.0938 |
| WEB | **EDGE** | **0.0132** | **0.1062** | 0.1094 | 0.1950 | COL | EDGE | 0.0042 | **0.2161** | 0.1325 | **0.3049** |
| | LGGM | 0.1225 | 0.1283 | **0.0976** | **0.1840** | | **LGGM** | **0.0026** | 0.3058 | **0.1285** | 0.3104 |
| ROAD | EDGE | 0.0254 | 0.1314 | 0.1313 | 0.1065 | ECO | EDGE | 0.0367 | 0.2424 | 0.0665 | **0.2156** |
| | **LGGM** | **0.0222** | **0.0624** | **0.1242** | **0.0867** | | **LGGM** | **0.0197** | **0.2406** | **0.0349** | **0.2156** |
| POWER | EDGE | 0.1417 | 0.2811 | 0.2568 | 0.4298 | CITATION | EDGE | 0.0124 | 0.0962 | 0.0460 | 0.0438 |
| | **LGGM** | **0.1276** | **0.2276** | **0.2548** | **0.3549** | | **LGGM** | **0.0073** | **0.0947** | **0.0448** | **0.0458** |

## F.6 TEXT-TO-GRAPH GENERATION

### F.6.1 DOMAIN SPECIFIC TRANSITION

Table 16: Comparing the performance of graph generation between LGGM trained on graphs from all domains with and without domain/name as textual conditions.

| Domain | Method | DEG | CC | Spec | Orb | Domain | Method | DEG | CC | Spec | Orb |
|---|---|---|---|---|---|---|---|---|---|---|---|
| FB | LGGM | 0.2566 | 0.3552 | 0.0587 | 0.1614 | BIO | LGGM | 0.2860 | 0.3275 | 0.1117 | 0.2333 |
| | LGGM-T2G$^D$ | 0.1533 | 0.1894 | 0.0817 | 0.0492 | | LGGM-T2G$^D$ | 0.1313 | 0.5111 | 0.1340 | 0.3736 |
| | LGGM-T2G$^{UP}$ | **0.0053** | **0.0576** | **0.0076** | **0.0245** | | LGGM-T2G$^{UP}$ | **0.0219** | **0.0251** | **0.0126** | **0.0190** |
| ASN | LGGM | 0.1477 | 0.3003 | 0.1551 | 0.3719 | ECON | LGGM | 0.3540 | 0.3404 | 0.2078 | 0.2740 |
| | LGGM-T2G$^D$ | 0.0429 | 0.4742 | 0.0949 | 0.0401 | | LGGM-T2G$^D$ | 0.2346 | 0.1572 | 0.1550 | **0.0579** |
| | LGGM-T2G$^{UP}$ | **0.0161** | **0.1312** | **0.0344** | **0.0174** | | LGGM-T2G$^{UP}$ | **0.0869** | **0.0601** | **0.0412** | 0.0592 |
| EMAIL | LGGM | 0.1957 | 0.2629 | 0.0646 | 0.2118 | RT | LGGM | 0.4355 | 0.3924 | 0.4329 | 0.4966 |
| | LGGM-T2G$^D$ | 0.0874 | 0.3238 | 0.1472 | 0.2869 | | LGGM-T2G$^D$ | 0.0050 | 0.0940 | 0.0415 | 0.2870 |
| | LGGM-T2G$^{UP}$ | **0.0077** | **0.0316** | **0.0176** | **0.0365** | | LGGM-T2G$^{UP}$ | **0.0034** | **0.0253** | **0.0225** | **0.0869** |
| WEB | LGGM | 0.2461 | 0.3570 | 0.1853 | 0.4832 | COL | LGGM | 0.2616 | **0.3398** | 0.2305 | 0.7090 |
| | LGGM-T2G$^D$ | 0.1253 | 0.9088 | 0.1156 | 0.3884 | | LGGM-T2G$^D$ | 0.1301 | 0.9384 | 0.1963 | 0.2032 |
| | LGGM-T2G$^{UP}$ | **0.0771** | **0.2720** | **0.0732** | **0.1251** | | LGGM-T2G$^{UP}$ | **0.0845** | 0.5070 | **0.1378** | **0.1531** |
| ROAD | LGGM | 0.4315 | 0.8107 | 0.3192 | 0.6976 | ECO | LGGM | 0.4611 | 0.3108 | 0.1932 | 0.3468 |
| | LGGM-T2G$^D$ | 0.0112 | 0.1611 | **0.0298** | 0.0120 | | LGGM-T2G$^D$ | **0.0575** | 0.2976 | **0.0585** | 0.2580 |
| | LGGM-T2G$^{UP}$ | **0.0097** | **0.1316** | 0.0324 | **0.0119** | | LGGM-T2G$^{UP}$ | 0.1070 | **0.2913** | 0.0410 | **0.2556** |
| POWER | LGGM | 0.4411 | **0.4694** | 0.3384 | 1.3222 | CITATION | LGGM | 0.3392 | 0.5009 | 0.1295 | 0.2248 |
| | LGGM-T2G$^D$ | **0.0194** | 0.6031 | **0.0286** | **0.0193** | | LGGM-T2G$^D$ | 0.1636 | 0.8868 | 0.2036 | 0.6142 |
| | LGGM-T2G$^{UP}$ | 0.0227 | 0.4817 | 0.0330 | 0.0223 | | LGGM-T2G$^{UP}$ | **0.0496** | **0.0914** | **0.0669** | **0.0318** |
| ALL | LGGM | 0.3213 | 0.3973 | 0.2022 | 0.4610 | | | | | | |
| | LGGM-T2G$^D$ | 0.0968 | 0.4621 | 0.1072 | 0.2158 | | | | | | |
| | LGGM-T2G$^{UP}$ | **0.0410** | **0.1755** | **0.0434** | **0.0703** | | | | | | |

### F.6.2 UNIFORM TRANSITION

Table 17: Comparing the performance of graph generation between LGGM trained on graphs from all domains with and without domain/name as textual conditions.

| Domain | Method | DEG | CC | Spec | Orb | Domain | Method | DEG | CC | Spec | Orb |
|---|---|---|---|---|---|---|---|---|---|---|---|
| FB | LGGM | 0.0321 | 0.4994 | 0.0763 | 0.3117 | BIO | LGGM | 0.2661 | 0.3120 | 0.1135 | 0.3835 |
| | LGGM-T2G$^D$ | 0.1561 | 0.1639 | 0.0924 | 0.0417 | | LGGM-T2G$^D$ | 0.0099 | 0.1286 | 0.0303 | 0.1366 |
| | LGGM-T2G$^{UP}$ | **0.0050** | **0.0545** | **0.0070** | **0.0251** | | LGGM-T2G$^{UP}$ | **0.0028** | **0.0287** | **0.0236** | **0.0174** |
| ASN | LGGM | 0.1511 | 0.4325 | 0.1875 | 0.3896 | ECON | LGGM | 0.3828 | 0.1533 | 0.2039 | 0.2583 |
| | LGGM-T2G$^D$ | 0.0318 | 0.2821 | 0.0606 | 0.0631 | | LGGM-T2G$^D$ | 0.0666 | 0.0594 | 0.0650 | 0.0586 |
| | LGGM-T2G$^{UP}$ | **0.0211** | **0.1191** | **0.0462** | **0.0195** | | LGGM-T2G$^{UP}$ | **0.0132** | **0.0257** | **0.0053** | **0.0191** |
| EMAIL | LGGM | 0.2156 | 0.2450 | 0.0666 | 0.2757 | RT | LGGM | 0.4395 | 0.2225 | 0.4337 | 0.6641 |
| | LGGM-T2G$^D$ | 0.0469 | 0.0982 | 0.0484 | 0.0505 | | LGGM-T2G$^D$ | 0.0468 | 0.0955 | 0.0729 | 0.0393 |
| | LGGM-T2G$^{UP}$ | **0.0073** | **0.0379** | **0.0127** | **0.0437** | | LGGM-T2G$^{UP}$ | **0.0286** | **0.0933** | **0.0400** | **0.0312** |
| WEB | LGGM | 0.2725 | 0.2672 | 0.1900 | 0.4368 | COL | LGGM | 0.3565 | 0.3554 | 0.2451 | 0.7874 |
| | LGGM-T2G$^D$ | 0.0255 | **0.0737** | 0.0354 | 0.1856 | | LGGM-T2G$^D$ | 0.0395 | 0.3110 | 0.1146 | 0.1823 |
| | LGGM-T2G$^{UP}$ | **0.0105** | 0.0941 | **0.0206** | **0.0451** | | LGGM-T2G$^{UP}$ | **0.0265** | **0.2813** | **0.0895** | **0.0899** |
| ROAD | LGGM | 0.4825 | 0.5373 | 0.3398 | 0.7542 | ECO | LGGM | 0.5466 | 0.6003 | 0.2257 | 0.7089 |
| | LGGM-T2G$^D$ | **0.0088** | 0.1225 | 0.0399 | 0.0155 | | LGGM-T2G$^D$ | 0.2160 | 0.2917 | 0.1203 | 0.2569 |
| | LGGM-T2G$^{UP}$ | 0.0177 | **0.0437** | **0.0336** | **0.0086** | | LGGM-T2G$^{UP}$ | **0.0293** | **0.2885** | **0.0416** | **0.2556** |
| POWER | LGGM | 0.4394 | 0.4646 | 0.3473 | 1.3186 | CITATION | LGGM | 0.2624 | 0.5374 | 0.1295 | 0.3419 |
| | LGGM-T2G$^D$ | 0.0162 | 0.1131 | 0.0479 | 0.1786 | | LGGM-T2G$^D$ | 0.0101 | 0.1025 | 0.0315 | 0.0651 |
| | LGGM-T2G$^{UP}$ | **0.0062** | **0.0570** | **0.0111** | **0.0084** | | LGGM-T2G$^{UP}$ | **0.0072** | **0.0849** | **0.0115** | **0.0287** |
| ALL | LGGM | 0.3206 | 0.3856 | 0.2132 | 0.5526 | | | | | | |
| | LGGM-T2G$^D$ | 0.0562 | 0.1535 | 0.0633 | 0.1061 | | | | | | |
| | LGGM-T2G$^{UP}$ | **0.0146** | **0.1007** | **0.0286** | **0.0494** | | | | | | |

### F.7 SENSITIVE ANALYSIS ON NUMBER OF TRAINING DATA UNDER DOMAIN SPECIFIC TRANSITION

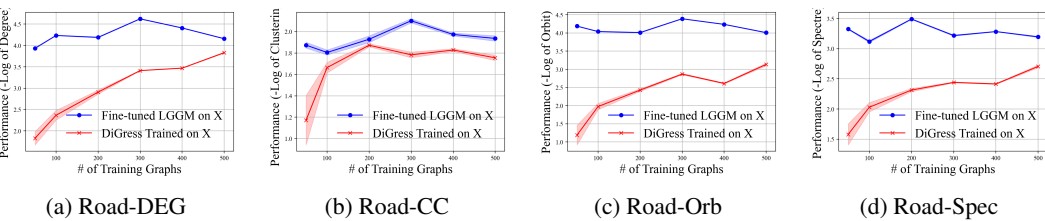

(a) Road-DEG     (b) Road-CC     (c) Road-Orb     (d) Road-Spec

Figure 9: Effect of Number of Training Graphs on Road Networks.

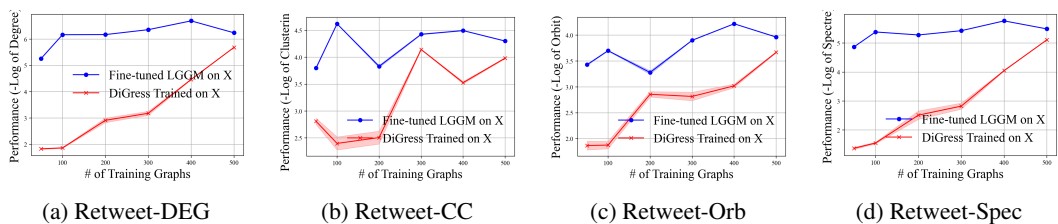

(a) Retweet-DEG     (b) Retweet-CC     (c) Retweet-Orb     (d) Retweet-Spec

Figure 10: Effect of Number of Training Graphs on Retweet Networks.

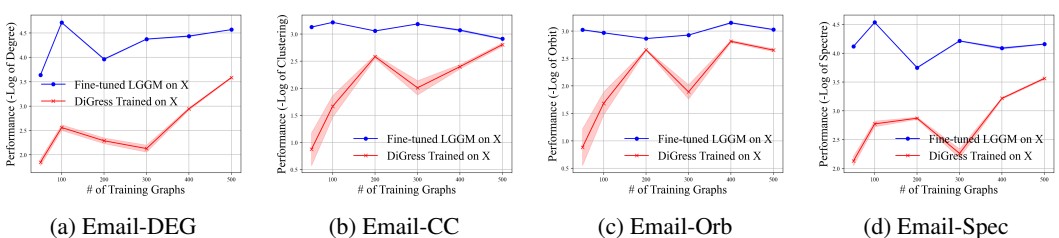

(a) Email-DEG     (b) Email-CC     (c) Email-Orb     (d) Email-Spec

Figure 11: Effect of Number of Training Graphs on Email Networks.

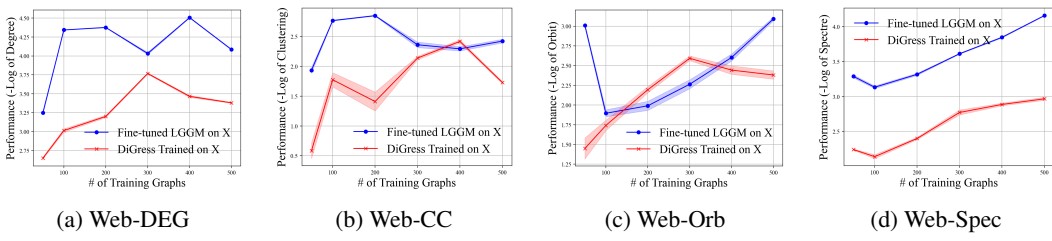

(a) Web-DEG     (b) Web-CC     (c) Web-Orb     (d) Web-Spec

Figure 12: Effect of Number of Training Graphs on Web Graphs.

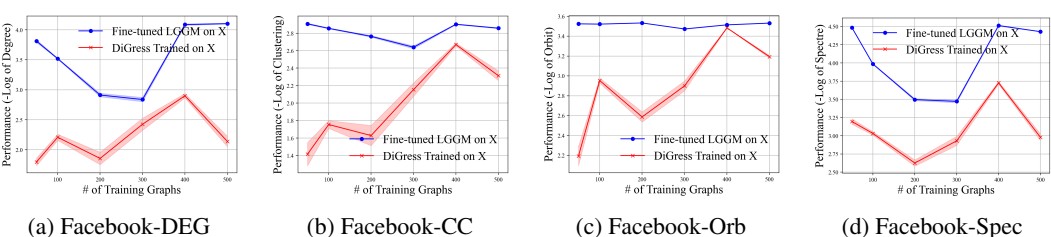

(a) Facebook-DEG     (b) Facebook-CC     (c) Facebook-Orb     (d) Facebook-Spec

Figure 13: Effect of Number of Training Graphs on Facebook Networks.

### F.8 SENSITIVE ANALYSIS ON NUMBER OF TRAINING DATA UNDER UNIFORM TRANSITION STRATEGY

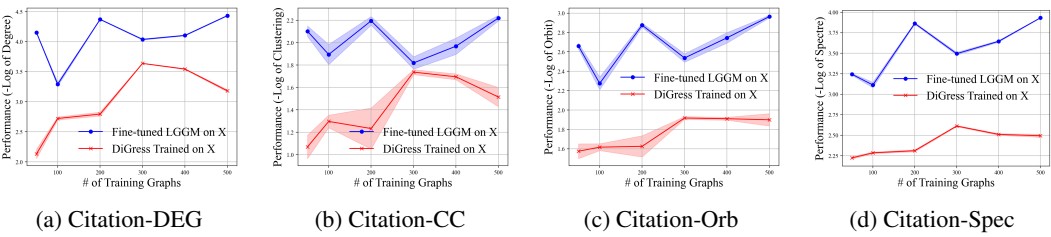

(a) Citation-DEG     (b) Citation-CC     (c) Citation-Orb     (d) Citation-Spec

Figure 14: Effect of Number of Training Graphs on Citation Networks.

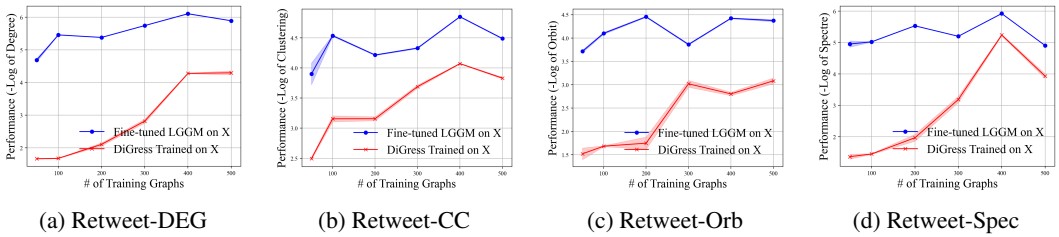

(a) Retweet-DEG     (b) Retweet-CC     (c) Retweet-Orb     (d) Retweet-Spec

Figure 15: Effect of Number of Training Graphs on Retweet Networks.

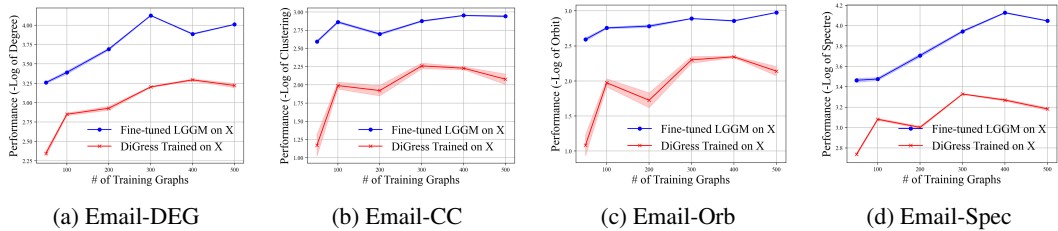

(a) Email-DEG     (b) Email-CC     (c) Email-Orb     (d) Email-Spec

Figure 16: Effect of Number of Training Graphs on Email Networks.

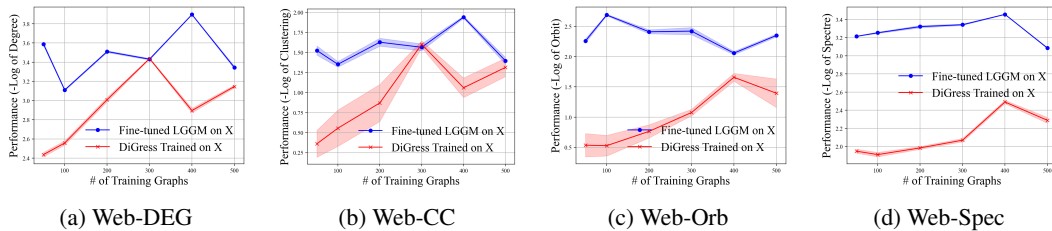

(a) Web-DEG     (b) Web-CC     (c) Web-Orb     (d) Web-Spec

Figure 17: Effect of Number of Training Graphs on Web Graphs.

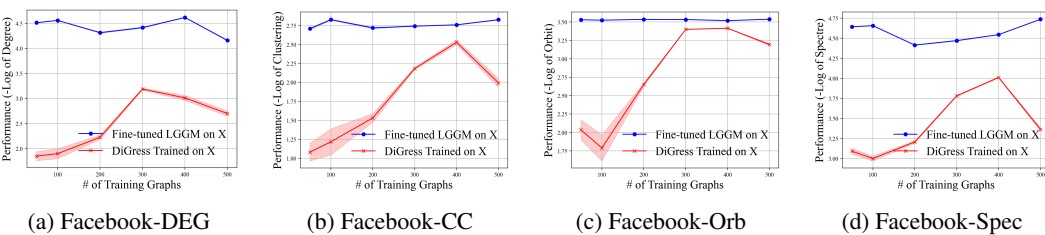

(a) Facebook-DEG     (b) Facebook-CC     (c) Facebook-Orb     (d) Facebook-Spec

Figure 18: Effect of Number of Training Graphs on Facebook Networks.

## F.9 COMPARING THE DOMAIN AS THE TEXTUAL CONDITION BEFORE/AFTER SHUFFLING

### F.9.1 DOMAIN SPECIFIC TRANSITION

Table 18: Comparing the performance of graph generation between LGGM trained on graphs from all domains with and without domain/name as textual conditions under domain-specific transition.

| Domain | Method | DEG | CC | Spec | Orb | Domain | Method | DEG | CC | Spec | Orb |
|---|---|---|---|---|---|---|---|---|---|---|---|
| FB | LGGM-T2G$^D$ | **0.1533** | **0.1894** | **0.0817** | **0.0492** | BIO | LGGM-T2G$^D$ | **0.1313** | **0.5111** | **0.1340** | **0.3736** |
|  | LGGM-T2G$^D$* | 0.2323 | 0.2618 | 0.1590 | 0.0923 |  | LGGM-T2G$^D$* | 0.1762 | 0.5887 | 0.1460 | 0.4929 |
| ASN | LGGM-T2G$^D$ | **0.0429** | **0.4742** | **0.0949** | **0.0401** | ECON | LGGM-T2G$^D$ | 0.2346 | **0.1572** | **0.1550** | **0.0579** |
|  | LGGM-T2G$^D$* | 0.0891 | 0.5725 | 0.1446 | 0.0610 |  | LGGM-T2G$^D$* | **0.2029** | 0.3393 | 0.2298 | 0.0579 |
| EMAIL | LGGM-T2G$^D$ | **0.0874** | **0.3238** | **0.1472** | **0.2869** | RT | LGGM-T2G$^D$ | **0.0050** | **0.0940** | **0.0415** | **0.2870** |
|  | LGGM-T2G$^D$* | 0.2169 | 0.7497 | 0.2825 | 0.8397 |  | LGGM-T2G$^D$* | 0.0240 | 0.1023 | 0.1374 | 0.4123 |
| WEB | LGGM-T2G$^D$ | **0.1253** | **0.9088** | **0.1156** | **0.3884** | COL | LGGM-T2G$^D$ | **0.1301** | **0.9384** | **0.1963** | **0.2032** |
|  | LGGM-T2G$^D$* | 0.1464 | 0.9776 | 0.1460 | 0.4211 |  | LGGM-T2G$^D$* | 0.1529 | 0.9684 | 0.2313 | 0.2089 |
| ROAD | LGGM-T2G$^D$ | **0.0112** | **0.1611** | **0.0298** | **0.0120** | ECO | LGGM-T2G$^D$ | **0.0575** | **0.2976** | **0.0585** | 0.2580 |
|  | LGGM-T2G$^D$* | 0.0365 | 0.2430 | 0.0605 | 0.0500 |  | LGGM-T2G$^D$* | 0.1964 | 0.3330 | 0.1438 | **0.2574** |
| POWER | LGGM-T2G$^D$ | **0.0194** | **0.6031** | **0.0286** | **0.0193** | CITATION | LGGM-T2G$^D$ | 0.1636 | **0.8868** | 0.2036 | 0.6142 |
|  | LGGM-T2G$^D$* | 0.0434 | 0.6721 | 0.0626 | 0.0231 |  | LGGM-T2G$^D$* | **0.1615** | 0.9553 | **0.1903** | **0.6078** |
| ALL | LGGM-T2G$^D$ | **0.0968** | **0.4621** | **0.1072** | **0.2158** |  |  |  |  |  |  |
|  | LGGM-T2G$^D$* | 0.1399 | 0.5636 | 0.1611 | 0.2937 |  |  |  |  |  |  |

### F.9.2 UNIFORM TRANSITION

Table 19: Comparing the performance of graph generation between LGGM trained on graphs from all domains with and without domain/name as textual conditions under uniform transition strategy.

| Domain | Method | DEG | CC | Spec | Orb | Domain | Method | DEG | CC | Spec | Orb |
|---|---|---|---|---|---|---|---|---|---|---|---|
| FB | LGGM-T2G$^D$ | **0.1561** | **0.1639** | **0.0924** | **0.0417** | BIO | LGGM-T2G$^D$ | **0.0099** | **0.1286** | **0.0303** | **0.1366** |
|  | LGGM-T2G$^D$* | 0.3018 | 0.4207 | 0.2069 | 0.2622 |  | LGGM-T2G$^D$* | 0.0754 | 0.2889 | 0.0881 | 0.2783 |
| ASN | LGGM-T2G$^D$ | **0.0318** | 0.2821 | **0.0606** | **0.0631** | ECON | LGGM-T2G$^D$ | **0.0665** | **0.0594** | **0.0650** | **0.0586** |
|  | LGGM-T2G$^D$* | 0.0637 | **0.1561** | 0.1416 | 0.2351 |  | LGGM-T2G$^D$* | 0.1035 | 0.0736 | 0.0971 | 0.0922 |
| EMAIL | LGGM-T2G$^D$ | **0.0469** | **0.0982** | **0.0484** | **0.0505** | RT | LGGM-T2G$^D$ | **0.0468** | **0.0955** | **0.0729** | **0.0393** |
|  | LGGM-T2G$^D$* | 0.1107 | 0.2322 | 0.1315 | 0.1692 |  | LGGM-T2G$^D$* | 0.1399 | 0.3913 | 0.2441 | 0.2497 |
| WEB | LGGM-T2G$^D$ | **0.0255** | **0.0737** | **0.0354** | **0.1856** | COL | LGGM-T2G$^D$ | 0.0395 | **0.3110** | **0.1146** | **0.1823** |
|  | LGGM-T2G$^D$* | 0.0485 | 0.0830 | 0.1340 | 0.2669 |  | LGGM-T2G$^D$* | **0.0323** | 0.4972 | 0.1159 | 0.5375 |
| ROAD | LGGM-T2G$^D$ | **0.0088** | 0.1225 | **0.0399** | **0.0155** | ECO | LGGM-T2G$^D$ | **0.2160** | **0.2917** | **0.1203** | **0.2569** |
|  | LGGM-T2G$^D$* | 0.0453 | **0.1005** | 0.1257 | 0.3803 |  | LGGM-T2G$^D$* | 0.3722 | 0.3210 | 0.2226 | 0.2771 |
| POWER | LGGM-T2G$^D$ | **0.0162** | **0.1131** | **0.0479** | **0.1786** | CITATION | LGGM-T2G$^D$ | **0.0101** | **0.1025** | **0.0315** | **0.0651** |
|  | LGGM-T2G$^D$* | 0.0225 | 0.1533 | 0.1264 | 0.2957 |  | LGGM-T2G$^D$* | 0.0375 | 0.2454 | 0.0699 | 0.1363 |
| ALL | LGGM-T2G$^D$ | **0.0562** | **0.1535** | **0.0633** | **0.1061** |  |  |  |  |  |  |
|  | LGGM-T2G$^D$* | 0.1128 | 0.2469 | 0.1420 | 0.2650 |  |  |  |  |  |  |

