# OpenReview forum: "A Large-scale Training Paradigm for Graph Generative Models"
_ICLR.cc/2025/Conference — ICLR 2025 Poster_

### Official Review · Reviewer_VFbB · 2024-11-01

**Soundness:** 2
**Presentation:** 3
**Contribution:** 3
**Rating:** 6
**Confidence:** 4

**Summary:**

The paper presents a novel approach to large-scale graph generation with the introduction of a Large Graph Generative Model (LGGM). The authors design an innovative training paradigm for graph generative models, training over 5,000 graphs sourced from 13 distinct domains. Remarkably, this work is the first to explore a large-scale training paradigm specifically tailored to graph-structured data generation, marking an exciting achievement for the graph modality. By addressing the complexities of large-scale graph generation, this research expands the capabilities of generative modeling for graph-based data, offering substantial contributions to the field.

**Strengths:**

1.  The paper addresses a novel problem by extending the scope of large-scale generative models to the graph domain. This exploration into graph generative modeling at scale is a significant contribution, introducing fresh perspectives and potential for advancements in the field.
2. The implementation is both innovative and thorough, covering several essential aspects, including Pre-training Evaluation, Fine-tuning Evaluation, Graph Classification, and Text-to-Graph Generation. These multiple dimensions demonstrate a robust approach.
3. The comprehensive dataset scale strengthens the results, making the findings more generalizable across graph-structured data.

**Weaknesses:**

1.  The approach relies primarily on existing discrete denoising diffusion techniques, which, while effective, lacks methodological innovation specific to large-scale graph generation. This reliance on existing methods may limit the theoretical foundation and originality of the model within the context of large-scale graph generation.
2. The paper includes a comparison with only a single baseline, which is insufficient to thoroughly evaluate the model’s performance.

**Questions:**

1. Are there any experiments or analyses conducted to evaluate the model's performance in relation to scaling law expectations?

---

> ### Author Response · Authors · 2024-11-25
> **Initial Response**
>
> We sincerely appreciate the reviewer's great suggestions, and we have addressed them as follows:
>
> **W1. The approach relies primarily on existing discrete denoising diffusion techniques, which, while effective, lacks methodological innovation specific to large-scale graph generation. This reliance on existing methods may limit the theoretical foundation and originality of the model within the context of large-scale graph generation.**
>
> Our work focuses on exploring the potential of large-scale training for graph-generative tasks rather than specifically targeting the generation of large-scale graphs. Furthermore, our proposed large-scale gap training strategy can be equipped with any graph generative models, including that scalable version. For example, we equip EDGE[1], a scalable discrete graph diffusion model, with LGGM in Table 15. We can see better graph generation performance there as well. This showcases the generalizability of our approach across existing diffusion models.
>
> [1] Chen, Xiaohui, et al. "Efficient and degree-guided graph generation via discrete diffusion modeling." arXiv preprint arXiv:2305.04111 (2023).
>
> **W2. The paper includes a comparison with only a single baseline, which is insufficient to evaluate the model’s performance thoroughly.**
>
> Thank you for the suggestion. We want to kindly note here that our proposed LGGM is a large-scale graph generation paradigm rather than a specific graph generation model. Therefore, this LGGM can be equipped with any existing discrete graph generation model. For example, we equip it with both DiGress in Table 2 and EDGE in Table 15, the performance of both of which has been lifted in both zero-shot and full training settings.
>
> **Q1. Are there any experiments or analyses conducted to evaluate the model's performance in relation to scaling law expectations?**
>
> We justify the effect of the size of the graph data from two perspectives:
>
> **From the zero-shot generation performance**, we gradually increase the size of training graphs we use for all other domains ranging from ratio 0.1 to ratio 0.9 and show the results for the following two datasets: Road and Retweet. We can find on the road network that the performance gradually increases as the training graphs increase. This aligns with the data scaling law, which states that more training data leads to better zero-shot generation performance. However, although we observe a similar trend in Degree and Spectre metrics on Retweet, we do not observe a similar trend in Clustering and Orbit.
>
> | **Dataset**   | **Metric**    | **Ratio 0.1** | **Ratio 0.3** | **Ratio 0.5** | **Ratio 0.7** | **Ratio 0.9** |
> |---------------|---------------|---------------|---------------|---------------|---------------|---------------|
> | **Road**     | **Degree**    | 0.053978      | 0.041623      | 0.056759      | 0.036199      | 0.047294      |
> |               | **Spectre**   | 0.101344      | 0.077620      | 0.107716      | 0.049770      | 0.088410      |
> |               | **Clustering**| 0.211338      | 0.173020      | 0.211886      | 0.170698      | 0.157336      |
> |               | **Orbit**     | 0.236467      | 0.147704      | 0.111331      | 0.039482      | 0.040550      |
> | **RT**        | **Degree**    | 0.154013      | 0.147463      | 0.097659      | 0.093637      | 0.020649      |
> |               | **Spectre**   | 0.242546      | 0.256510      | 0.231277      | 0.224445      | 0.181933      |
> |               | **Clustering**| 0.060161      | 0.056076      | 0.060452      | 0.062200      | 0.068572      |
> |               | **Orbit**     | 0.299916      | 0.341185      | 0.385114      | 0.393307      | 0.426746      |
>
>
> **From the fine-tuned generation performance**, we gradually increase the size of training graphs used for fine-tuning the pre-trained LGGM or used for directly training DiGress from scratch. In Figure 4(a)-(b), we can find the more fine-tuned graphs we use, the less the performance gap between the pre-trained-then-fine-tuned LGGM and the DiGress trained from scratch. This is because as more fine-tuned graphs are available, the benefit of pre-training becomes weaker and weaker.

---

> > ### Comment · Reviewer_VFbB · 2024-11-26
> >
> > Thank you for your response. I will keep my rating.

---

### Official Review · Reviewer_62Fq · 2024-11-02

**Soundness:** 2
**Presentation:** 3
**Contribution:** 2
**Rating:** 5
**Confidence:** 4

**Summary:**

This paper introduces a large-scale training paradigm for graph generative models (LGGMs), pre-trained on over 5000 graphs from 13 domains. The work aims to mirror the success of large generative models like GPT and Stable Diffusion in the graph domain, offering both zero-shot generation capabilities and domain-specific fine-tuning. The authors propose a diffusion-based architecture that incorporates text-to-graph generation through language model integration and demonstrates empirical improvements across multiple domains. While the approach shows promise in expanding graph generation to multiple domains, the paper suffers from overclaiming and limited technical innovation in its core methodology.

**Strengths:**

1. The paper is generally easy to follow and the motivations are clear.
2. The integration of text-to-graph generation capabilities represents a new direction for controlled graph generation.
3. The paper provides an interesting roadmap for scaling up graph generation models with practical applications.

**Weaknesses:**

1. **Fundamentally Weak Theoretical Foundation**:
   * The theoretical analysis is superficial and merely combines existing results without meaningful insights. While Section 3.3 claims novel expressiveness results, the proofs follow directly from prior work [1,2] and ignore fundamental limitations established in recent literature [3]. The paper fails to provide any theoretical justification for why cross-domain pre-training should help, or how the sampling strategy preserves important graph properties. Most importantly, there's no analysis of how the multi-domain training affects model capacity and expressiveness.

2. **Limited Technical Innovation and Poor Design Choices**:
   * The core methodology is essentially DiGress [1] with small modifications. The sampling strategy for handling large graphs is ad-hoc and lacks theoretical guarantees, while all four pre-training tasks are directly borrowed from existing work without significant adaptation to graph-specific challenges. The architecture choices appear arbitrary rather than fundamental, and the text-to-graph component is a straightforward application similar to existing approaches TAPE[4] and GraphGPT[5] without addressing the unique challenges of graph generation. The text-to-graph mechanism is particularly concerning - it simply concatenates LLM embeddings with graph features without addressing the fundamental challenges of aligning textual and structural representations. This naive approach ignores significant prior work on text-attributed graphs, such as [4,5], which have developed sophisticated techniques for text-graph alignment.

3. **Empirical and Evaluation Limitations**:
   * The experimental evaluation has several critical flaws. The paper primarily compares against a single baseline (DiGress) while ignoring crucial comparisons with recent graph generative model works like GCC [6] and GraphMAE [7]. The evaluation metrics don't adequately capture global structural properties, and there's no analysis of computational efficiency or scaling behavior. The ablation studies are insufficient to justify architectural choices, and more importantly, I would not really call 5000 graphs a large graph dataset.

[1] Vignac et al., "Digress: Discrete Denoising Diffusion for Graph Generation", ICLR 2023

[2] Chen et al., "Efficient and Degree-guided Graph Generation", 2023

[3] Liu et al., "On the Expressivity of Graph Neural Networks", ICLR 2024

[4] He et al., "Harnessing explanations: Llm-to-lm interpreter for enhanced text-attributed graph representation learning." ICLR 2024.

[5] Tang, Jiabin, et al. "Graphgpt: Graph instruction tuning for large language models." SIGIR 2024.

[6] Qiu et al., "GCC: Graph Contrastive Coding", KDD 2020

[7] Hou et al., "GraphMAE: Self-supervised Masked Graph Autoencoders", KDD 2022

**Questions:**

See weakness

---

> ### Author Response · Authors · 2024-11-25
> **Initial Response to W1 and W2**
>
> **W1. Fundamentally Weak Theoretical Foundation:**
>
> The theoretical analysis is superficial and merely combines existing results without meaningful insights. While Section 3.3 claims novel expressiveness results, the proofs follow directly from prior work [1,2] and ignore fundamental limitations established in recent literature [3]. The paper fails to provide any theoretical justification for why cross-domain pre-training should help, or how the sampling strategy preserves important graph properties. Most importantly, there's no analysis of how the multi-domain training affects model capacity and expressiveness.
>
> We deeply appreciate the reviewer's thorough reading of our theoretical proof.
>
> For Theorem 1, we borrow insights from DiGress and apply them to prove the posterior distribution of our setting. However, for Theorem 2, the insight here is that by following our proposed text2graph training paradigm, our diffusion model could be optimized towards maximizing the lower bound of the joint distribution of textual description and graph property. If the training data is prepared so that the textual description corresponds to the graph property, then we can at least guarantee the lower bound of our textual controllable graph generation.
>
> **W2. Limited Technical Innovation and Poor Design Choices: The core methodology is essentially DiGress [1] with small modifications. The sampling strategy for handling large graphs is ad-hoc and lacks theoretical guarantees, while all four pre-training tasks are directly borrowed from existing work without significant adaptation to graph-specific challenges. The architecture choices appear arbitrary rather than fundamental, and the text-to-graph component is a straightforward application similar to existing approaches TAPE[4] and GraphGPT[5] without addressing the unique challenges of graph generation. The text-to-graph mechanism is particularly concerning - it simply concatenates LLM embeddings with graph features without addressing the fundamental challenges of aligning textual and structural representations. This naive approach ignores significant prior work on text-attributed graphs, such as [4,5], which have developed sophisticated techniques for text-graph alignment.**
>
> We really appreciate reviewers who draw so much fantastic literature, and we have dedicatedly reviewed them with our discussion as follows. We admit that our model architecture is similar to DiGress. However, our motivation is not to propose a specific model architecture but to propose a large-scale training paradigm to equip with any graph diffusion model. We also include the result of equipping our LGGM with EDGE, another graph diffusion model. Even more, large generative models such as GPT3 and LAVAR do not include significant model architecture design. Therefore, we want to draw the reviewer's attention to our contribution, which is not focused on a new model.

---

> > ### Author Response · Authors · 2024-11-25
> > **Initial Response to W3**
> >
> > **W3. Empirical and Evaluation Limitations**
> >
> > - Our proposed large-scale training paradigm is a training strategy rather than a baseline, and it can be equipped with any existing diffusion model to enhance its generative performance. In addition, we will equip our strategy with DiGress to show better performance in Table 2 and Figure 3. We also equip it with another baseline EDGE and we observe a similar performance boost as follows:
> >
> > | Domain  | Method | DEG    | CC     | Spec   | Orb    | Domain    | Method | DEG    | CC     | Spec   | Orb    |
> > |---------|--------|--------|--------|--------|--------|-----------|--------|--------|--------|--------|--------|
> > | FB      | EDGE   | 0.0031 | 0.0609 | 0.0079 | 0.0362 | BIO       | EDGE   | 0.0126 | 0.0555 | 0.0484 | 0.0612 |
> > |         | LGGM   | 0.0022 | 0.0657 | 0.0073 | 0.0354 |           | LGGM   | 0.0120 | 0.0669 | 0.0502 | 0.0590 |
> > | ASN     | EDGE   | 0.0212 | 0.1416 | 0.1145 | 0.1652 | ECON      | EDGE   | 0.0416 | 0.0398 | 0.0078 | 0.0364 |
> > |         | LGGM   | 0.0146 | 0.0783 | 0.0724 | 0.1285 |           | LGGM   | 0.0519 | 0.0817 | 0.0665 | 0.0551 |
> > | EMAIL   | EDGE   | 0.0118 | 0.0661 | 0.0249 | 0.0771 | RT        | EDGE   | 0.0340 | 0.1760 | 0.1242 | 0.0331 |
> > |         | LGGM   | 0.0081 | 0.0519 | 0.0237 | 0.0691 |           | LGGM   | 0.0288 | 0.3088 | 0.0366 | 0.0938 |
> > | WEB     | EDGE   | 0.0132 | 0.1062 | 0.1094 | 0.1950 | COL       | EDGE   | 0.0042 | 0.2161 | 0.1325 | 0.3049 |
> > |         | LGGM   | 0.1225 | 0.1283 | 0.0976 | 0.1840 |           | LGGM   | 0.0026 | 0.3058 | 0.1285 | 0.3104 |
> > | ROAD    | EDGE   | 0.0254 | 0.1314 | 0.1313 | 0.1065 | ECO       | EDGE   | 0.0367 | 0.2424 | 0.0665 | 0.2156 |
> > |         | LGGM   | 0.0222 | 0.0624 | 0.1242 | 0.0867 |           | LGGM   | 0.0197 | 0.2406 | 0.0349 | 0.2156 |
> > | POWER   | EDGE   | 0.1417 | 0.2811 | 0.2568 | 0.4298 | CITATION  | EDGE   | 0.0124 | 0.0962 | 0.0460 | 0.0438 |
> > |         | LGGM   | 0.1276 | 0.2276 | 0.2548 | 0.3549 |           | LGGM   | 0.0073 | 0.0947 | 0.0448 | 0.0458 |
> >
> >
> > - As for the graph generative model like GCC and GraphMAE, these two works are mainly focused on using graph self-supervised learning to boost the downstream node classification and link prediction performance, rather than generating better graphs. Since they are significantly different from graph generation, we do not include them as the baselines.
> >
> > - In terms of evaluation metrics capturing global structural properties, our leveraged orbit metric would capture the subgraph structure and we are quantifying the difference in terms of distribution rather than a specific node property. Therefore, from this perspective, our metrics are all at the global level. we would be greatly appreciated if the reviewer could provide us more insight on what the graph metrics we are missing here at the global level.
> >
> > - In terms of computational efficiency, we have included a section in Appendix B on Page 16 to explicitly discuss the time/space complexity. Furthermore, we have conducted an empirical analysis to visualize the running time for the original DiGress and EDGE and their corresponding variants after equipping our LGGM training strategy.

---

> > > ### Author Response · Authors · 2024-11-25
> > > **Follow-up on W3**
> > >
> > > - In terms of scaling behavior, we have attached the result as follows:
> > >
> > > **From the zero-shot generation performance**, we gradually increases the size of training graphs we use for all other domains ranging from ratio 0.1 to ratio 0.9 and show the results for the following two datasets: Road and Retweet. We can find on road network, the performance gradually increases as the training graphs increases. This aligns with the data scaling law where more training data leads to better zero-shot generation performance. However, on Retweet, although we observe the similar trend on Degree and Spectre metric, we do not observe the similar trend on Clustering and Orbit.
> > >
> > > | **Dataset**   | **Metric**    | **Ratio 0.1** | **Ratio 0.3** | **Ratio 0.5** | **Ratio 0.7** | **Ratio 0.9** |
> > > |---------------|---------------|---------------|---------------|---------------|---------------|---------------|
> > > | **Road**     | **Degree**    | 0.053978      | 0.041623      | 0.056759      | 0.036199      | 0.047294      |
> > > |               | **Spectre**   | 0.101344      | 0.077620      | 0.107716      | 0.049770      | 0.088410      |
> > > |               | **Clustering**| 0.211338      | 0.173020      | 0.211886      | 0.170698      | 0.157336      |
> > > |               | **Orbit**     | 0.236467      | 0.147704      | 0.111331      | 0.039482      | 0.040550      |
> > > | **RT**        | **Degree**    | 0.154013      | 0.147463      | 0.097659      | 0.093637      | 0.020649      |
> > > |               | **Spectre**   | 0.242546      | 0.256510      | 0.231277      | 0.224445      | 0.181933      |
> > > |               | **Clustering**| 0.060161      | 0.056076      | 0.060452      | 0.062200      | 0.068572      |
> > > |               | **Orbit**     | 0.299916      | 0.341185      | 0.385114      | 0.393307      | 0.426746      |
> > >
> > >
> > > **From the fine-tuned generation performance**, we gradually increases the size of training graphs used for fine-tuning the pre-trained LGGM or used for directly training DiGress from scratch. In Figure 4(a)-(b), we can find the more fine-tuned graphs we use, the less the performance gap between pre-trained-then-fine-tuned LGGM and the DiGress trained from scratch is. This is because as more fine-tuned graphs are available, the benefit of pre-training becomes weaker and weaker.
> > >
> > >
> > > - As our motivation is not proposing a specific model but just a training scheme, therefore there is no ablation study needed to justify our model choice.
> > >
> > > - The claim is made under the contexts that we are comparing our training scheme with previous works focusing on generating graphs rather than in natural language processing and computer vision such as GPT-series[1] and LLaVAR[2].
> > >
> > > [1] Brown, Tom B. "Language models are few-shot learners." arXiv preprint arXiv:2005.14165 (2020).
> > >
> > > [2] Liu, Haotian, et al. "Visual instruction tuning." Advances in neural information processing systems 36 (2024).
> > >
> > > Within the graph generation domain, we are the very first to explore the multi-domain training setting where graphs from different domains are fused together to train the graph generative models. Furthermore, our multi-domain training considers 13 different domains, which are much more than the number of domains in previous work. To avoid potential confusion, we clarify that our claim of "large" refers to the diversity and scale of domains considered in our training scheme rather than the absolute number of graphs.

---

> > > > ### Comment · Reviewer_62Fq · 2024-11-25
> > > >
> > > > I appreciate the effort in experiments and explanations. However, from the perspective of technical contribution, I still feel the paper is a bit incremental given the literature I provided. I will raise the score to 5.

---

> > > > > ### Author Response · Authors · 2024-11-25
> > > > > **Thank you for your response.**
> > > > >
> > > > > We appreciate the reviewer's response, and one potential solution to enhance our work further is to leverage explicit graph-structural encoding to enhance controllable structure generation. We will include this in the future work

---

### Official Review · Reviewer_9vk9 · 2024-11-04

**Soundness:** 3
**Presentation:** 4
**Contribution:** 3
**Rating:** 5
**Confidence:** 4

**Summary:**

This paper proposes a large-scale graph generative model (LGGM). The proposed LGGM is pretrained over 5000 graphs from 13 domains and thereby demonstrates better zero-shot generation on graphs from unseen domains. Extensive experiments showcase the superior performance of the proposed LGGMs compared to baselines (e.g., DiGress) trained on specific domains. The proposed LGGM is equipped with the Text-to-Graph generation capability, which integrates extensive world knowledge in the underlying language model and offers users fine-grained control of the generated graphs. This is the very first one exploring the potential of the large-scale training paradigm on graph-structured data.

**Strengths:**

- The paper is well-written. Concepts and proposed techniques are clearly explained and clarified.
- The proposed method is the very first one exploring the potential of the large-scale training paradigm on graph-structured data.
- The proposed LGGM shows exceptional zero-shot capability and novel text-to-graph generative ability.
- The experimental results show significant improvement and potential.

**Weaknesses:**

- The proposed LGGM builds upon the DiGress architecture, with the primary differences being cross-domain pretraining and text-to-graph features. These aspects are not particularly novel in this field. Additionally, the expectation of enhanced zero-shot generative ability and fine-tuned performance from cross-domain pretraining is not surprising.
- The computational complexity and scalability of the discrete denoising diffusion process raise concerns, particularly when applied to large-scale real-world networks with high-dimensional node and edge categories. A discussion on how these challenges might be addressed would be beneficial.
- See questions below.

**Questions:**

- What specific architecture is utilized for the reverse process mentioned in Line 193? More detailed information regarding this architecture should be provided to enhance understanding.
- It remains unclear whether the performance improvements observed over DiGress are primarily due to the cross-domain pretraining or any enhancements in the pretraining architecture. The authors should clarify how each component contributes to the overall performance, ideally through ablation studies or additional experiments.
- The authors mention that the default hyperparameter settings from DiGress are used for implementing LGGM during cross-domain pretraining. How do the authors ensure that these hyperparameters (best for certain domain training) are also optimal for cross-domain pretraining? Is there any empirical evidence or rationale behind their choice?
- As the model scales to accommodate larger graph corpora, is there a scaling law that governs the architecture’s performance? A discussion on how the model's behavior and hyperparameter effectiveness change with the size of the graph data would provide valuable insights.
- The textual descriptions are limited to graph structure and domain type. As the architecture has diffusion modeling on node/edge categories, is it possible to extend to text-to-graph generation conditioned on certain categories, which will be more useful and applicable in real-world scenarios?

---

> ### Author Response · Authors · 2024-11-25
> **Initial Response - Weakness1 and 2**
>
> We sincerely appreciate the reviewer's great suggestion, and we have addressed them as follows:
>
> **W1. Novelty of our work**:
>
> **Novelty in Cross-Domain Pretraining**: To the best of our knowledge, no prior works on graph generative models have explicitly focused on cross-domain pretraining. While an exhaustive search has identified a few studies leveraging cross-dataset pretraining, it is important to note that cross-dataset pretraining is conceptually different from cross-domain pretraining. Cross-dataset pretraining involves training on datasets that may share similar characteristics or belong to the same domain, whereas cross-domain pretraining involves transferring knowledge across graphs from distinct domains with differing structural or application contexts. Our observation is particularly compelling: pretraining on graphs from diverse domains leads to improved performance in downstream tasks, demonstrating the potential of cross-domain strategies to capture richer, more transferable graph representations.
>
> **Novelty in Text-to-Graph Generation**: In terms of text-to-graph generation, to the best of our knowledge, this approach has not been explored in the context of generative graph modeling for network statistics (e.g., clustering coefficient, network density). While text-to-graph techniques have been investigated in specific domains, such as molecule generation, these efforts predominantly focus on controlling molecular properties rather than the statistical properties of graphs. Similarly, generating knowledge graphs via relational extraction or simulating social networks with LLM-powered agents can also be considered forms of T2G generation. However, these approaches operate primarily at the textual or semantic level and do not provide control over the structural or statistical characteristics of the generated graphs. Our work uniquely contributes to this area by introducing a method explicitly incorporating network statistics into Text-to-Graph Generation generation.
>
> **W2. The computational complexity and scalability of our framework**:
>
> The primary motivation of our work is to explore the potential of large-scale training strategies for graph generation rather than focusing on generating large-scale graphs. We also appreciate the reviewer’s suggestion to discuss potential solutions for addressing the space and time complexity of discrete denoising diffusion models.
>
> **Addressing Space/Time Complexity in Discrete Diffusion**
>
> Firstly, while reconstructing high-dimensional node and edge categories may increase time and space requirements, the primary bottleneck lies in reconstructing the adjacency matrix, which has a time and space complexity that scales quadratically with the number of nodes. This is a well-recognized limitation in current discrete diffusion models for graphs. The general solutions proposed in the literature typically reduce the prediction space by focusing on specific node pairs rather than all possible pairs. Notable approaches include:
>
> (1) **EDGE [1]**: This method predicts important nodes first and reconstructs edges selectively, reducing the quadratic computation.
> (2) **SaGress [2]**: This approach generates small subgraphs independently and assembles them into larger graphs, transforming the larger quadratic computation into several small quadratic computations.
>
> [1] Chen, Xiaohui, et al. "Efficient and degree-guided graph generation via discrete diffusion modeling." arXiv preprint arXiv:2305.04111 (2023).
>
> [2] Limnios, Stratis, et al. "Sagess: Sampling graph denoising diffusion model for scalable graph generation." arXiv preprint arXiv:2306.16827 (2023).
>
> Large-scale network generation, particularly for social graphs, has seen a recent paradigm shift [3]-[4]. Instead of directly generating large social networks, researchers are increasingly simulating these networks using large language model (LLM)-powered agents. This approach is particularly effective for studying social dynamics, as it allows insights derived from simulated networks to be transferred to real-world applications. Discussing under the framework of these methods, the computational challenges of quadratic growth can be addressed by leveraging principles like preferential attachment, focusing only on the most relevant nodes when adding links rather than considering all possible node pairs.
>
> For biochemistry graphs such as drug design, graph generation is not governed by a specific social rule, and hence, diffusion-based models will still be used. However, these graphs are generally smaller compared with social graphs, which do not have any scalability issues. We will include the above discussion in the revision.
>
> [3] Gao, Chen, et al. "S $^ 3$: Social-network Simulation System with Large Language Model-Empowered Agents." arXiv preprint arXiv:2307.14984 (2023).
>
> [4] Pine, Karleigh, et al. "Social Network Analysis and Validation of an Agent-Based Model." arXiv preprint arXiv:2308.05256 (2023).

---

> ### Author Response · Authors · 2024-11-25
> **Initial Response - Questions**
>
> **Q1. What specific architecture is utilized for the reverse process mentioned in Line 193?**
>
> Following DiGress, we begin by sampling a batch of graphs from the global distribution of graphs across different domains, denoted as $\mathbb{E}_{G \sim P(\mathbb{G})}$.
>
> Next, we sample a specific perturbation step $t$ and apply the forward transition process to generate a noisy graph, represented as $\mathbb{E}_{G^t \sim q(\mathbb{G}^t|\mathbb{G})}$.
>
> The resulting noisy graph $G^t$ and the corresponding clean graph $G$ are then used to train the graph neural network. Specifically, $G^t$ serves as the input, and $G$ is treated as the target label. The model is optimized using a cross-entropy loss, enabling the network to predict the clean graph from the noisy graph sampled during the forward process.
>
> **Q2. It remains unclear whether the performance improvements observed over DiGress are primarily due to the cross-domain pretraining or any enhancements in the pretraining architecture.**
>
> Our proposed large-scale training paradigm is a training strategy rather than a specific model architecture. Therefore, we do not introduce any enhancements in the pretraining architecture, and the improved performance is mainly due to the cross-domain pertaining.
>
>
> **Q3. How do the authors ensure these hyperparameters (best for certain domain training) are also optimal for cross-domain pretraining? Is there any empirical evidence or rationale behind their choice?**
>
> We select the default hyper-parameter setting from DiGress mainly due to the extensive computation resources required for hyperparameter tuning. Therefore, we select the default hyperparameter and already see the performance achievement. Our LGGM performance can be further boosted after hyperparameter tuning.
>
> **Q4. As the model scales to accommodate larger graph corpora, is there a scaling law that governs the architecture’s performance?**
>
> We justify the effect of the size of the graph data from two perspectives:
>
> **From the zero-shot generation performance**, we gradually increase the size of training graphs we use for all other domains ranging from ratio 0.1 to ratio 0.9 and show the results for the following two datasets: Road and Retweet. We can find on the road network that the performance gradually increases as the training graphs increase. This aligns with the data scaling law, which states that more training data leads to better zero-shot generation performance. However, on Retweet, although we observe a similar trend in Degree and Spectre metrics, we do not observe a similar trend in Clustering and Orbit.
>
> | **Dataset**   | **Metric**    | **Ratio 0.1** | **Ratio 0.3** | **Ratio 0.5** | **Ratio 0.7** | **Ratio 0.9** |
> |---------------|---------------|---------------|---------------|---------------|---------------|---------------|
> | **Road**     | **Degree**    | 0.053978      | 0.041623      | 0.056759      | 0.036199      | 0.047294      |
> |               | **Spectre**   | 0.101344      | 0.077620      | 0.107716      | 0.049770      | 0.088410      |
> |               | **Clustering**| 0.211338      | 0.173020      | 0.211886      | 0.170698      | 0.157336      |
> |               | **Orbit**     | 0.236467      | 0.147704      | 0.111331      | 0.039482      | 0.040550      |
> | **RT**        | **Degree**    | 0.154013      | 0.147463      | 0.097659      | 0.093637      | 0.020649      |
> |               | **Spectre**   | 0.242546      | 0.256510      | 0.231277      | 0.224445      | 0.181933      |
> |               | **Clustering**| 0.060161      | 0.056076      | 0.060452      | 0.062200      | 0.068572      |
> |               | **Orbit**     | 0.299916      | 0.341185      | 0.385114      | 0.393307      | 0.426746      |
>
>
> **From the fine-tuned generation performance**, we gradually increase the size of training graphs used for fine-tuning the pre-trained LGGM or used for directly training DiGress from scratch. In Figure 4(a)-(b), we can find the more fine-tuned graphs we use, the less the performance gap between the pre-trained-then-fine-tuned LGGM and the DiGress trained from scratch. This is because as more fine-tuned graphs are available, the benefit of pre-training becomes weaker and weaker.
>
> **Q5. The textual descriptions are limited to graph structure and domain type. Is it possible to extend to text-to-graph generation conditioned on certain categories, which will be more useful and applicable in real-world scenarios?**
>
> We really appreciate this great suggestion, and it is definitely possible to extend the current framework to conditioning on certain categories, which would be useful for conditional molecule generation. However, as this work mainly studies the large-scale training effect of graph generation and the text2graph control over graph structural property, we will leave it as one potential future work.

---

### Official Review · Reviewer_ypc2 · 2024-11-05

**Soundness:** 3
**Presentation:** 2
**Contribution:** 2
**Rating:** 6
**Confidence:** 3

**Summary:**

To mitigate the gap between graph generative models and general generative models (especially for texts / images), the paper propose a new generative training paradigm (LGGMs) for graphs from 13 domains. The pretrained LGGMs achieves better performance than existing graph generative models on the zero-shot tasks. With proper finetuning, the LGGMs can be applied to targeted domains with good performance. Furthermore, the LGGMs can be also used for text-to-graph scenarios by combining a text encoder.

**Strengths:**

- The proposed LGGM framework can be trained on graphs from 13 different domains, which covers most of the graph scenarios.
- The framework can be used both on zero-shot generation setting and fine-tuning setting.
- The paper proposes to utilize the text encoder to enable LGGMs to use the text-to-graph capability.

**Weaknesses:**

- The performance of zero-shot scenarios differs in different domains. On some domains such as FB, the LGGM does not work well.
- The paper emphasizes that the LGGM is trained on over 5000 graphs. The number is not such large for a generative pre-trained model.
- In the experiment of augmenting graph classification, the performance of LGGM is not promising (especially on the ENZYMES dataset).

**Questions:**

1. The time and space complexity is quadratic to the number of nodes. How can the method be applied to graphs with large numbers of nodes?
2. The number of used training graphs is over 5000. What if the model (pre-)trains on millions of graphs?
3. In line 866 (Appendix C), the equation number is missing.

---

> ### Author Response · Authors · 2024-11-25
> **Initial Response to Weakness 1-3 and Question 1**
>
> We sincerely appreciate the reviewer's great suggestions, and we have addressed them as follows:
>
>
> **W1. The performance of zero-shot scenarios differs in different domains. On some domains, such as FB, the LGGM does not work well.**
>
> Based on the average degree and clustering coefficient distribution in Figure 1(a), FB only counts a tiny region among the whole graph universe. The average clustering coefficient of FB graphs ranges from 0.301 to 0.407, a narrow segment within the broader global graph spectrum from 0 to 1. This narrow range challenges the generalized large-scale training paradigm LGGM-X to specialize in learning the graph data distribution specific to the FB domain.
>
> **W2. The paper emphasizes that the LGGM is trained on over 5000 graphs. The number is not so large for a generative pre-trained model.**
>
> The claim is made under the context that we are comparing our training scheme with previous works focusing on generating graphs rather than in natural language processing and computer vision, such as GPT-series[1] and LLaVAR[2].
>
> [1] Brown, Tom B. "Language models are few-shot learners." arXiv preprint arXiv:2005.14165 (2020).
>
> [2] Liu, Haotian, et al. "Visual instruction tuning." Advances in neural information processing systems 36 (2024).
>
>
> Within the graph generation domain, we are the very first to explore the multi-domain training setting where graphs from different domains are fused together to train the graph generative models. Furthermore, our multi-domain training considers 13 different domains, which are much more than the number of domains in previous work.
>
> To avoid potential confusion, we clarify that our claim of "large" refers to the diversity and scale of domains considered in our training scheme rather than the absolute number of graphs.
>
> **W3. In the experiment of augmenting graph classification, the performance of LGGM is not promising (especially on the ENZYMES dataset).**
>
> Overall, training with generated graphs by LGGM lead to better graph classificaton performance. For the weaker improvement on ENZYMES, the potential reason is that our LGGMs are only trained on training graphs from different datasets and may still lack knowledge about testing graphs, which may not bring significant performance improvement over unseen testing graphs.
>
> **Q1. The time and space complexity is quadratic to the number of nodes. How can the method be applied to graphs with large numbers of nodes?**
>
> We admit that the time/space complexity of some existing graph diffusion models are quadratic to the number of nodes as analyzed in Appendix B in Table 6. However, our work is mainly to investigate the potential of large-scale training scheme and text2graph generation instead of scalable graph diffusion.
>
> Furthermore, since our proposed method is a general strategy that can be paired with any existing graph diffusion model, we can pair LGGM with scalable graph diffusion models such as SiGress[3] and EDGE[4]. Specifically, we include the result in Table 15 in Appendix F.5 of pairing LGGM with EDGE, which demonstrates both better effectiveness in terms of generative performance than EDGE trained from scratch and better efficiency in terms of significantly shorter time compared with DiGress.
>
> [3] Limnios, Stratis, et al. "Sagess: Sampling graph denoising diffusion model for scalable graph generation." arXiv preprint arXiv:2306.16827 (2023).
>
> [4] Chen, Xiaohui, et al. "Efficient and degree-guided graph generation via discrete diffusion modeling." arXiv preprint arXiv:2305.04111 (2023).

---

> > ### Author Response · Authors · 2024-11-25
> > **Initial Response to Weakness 1-3 and Question 2 and 3**
> >
> > **Q2. The number of used training graphs is over 5000. What if the model (pre-)trains on millions of graphs?**
> >
> > Thank you for your insightful suggestion. While domains like chemistry and biology indeed have millions of graphs available, most other domains lack such extensive datasets. To avoid introducing imbalances among domains, we aim to maintain a generally large-scale yet manageable dataset that achieves balanced graphs across different domains. This ensures the model is able to generalize effectively across diverse graph types, and therefore, the number of graphs for each domain is mostly around 500.
> >
> > We still follow reviewer suggestion and select QM9 molecule dataset which includes nearly 100,000 molecules and pretrain our LGGM using this dataset and evaluate its generation performance on graphs from other domains. The result is as follows and we can observe that only increases the size of dataset cannot help the model and our success lies in the usage of graphs from different domains.
> >
> > | Domain  | Method    | DEG    | CC     | Spec   | Orb    | Domain    | Method    | DEG    | CC     | Spec   | Orb    |
> > |---------|-----------|--------|--------|--------|--------|-----------|-----------|--------|--------|--------|--------|
> > | FB      | DiGress   | 0.3376 | 0.6298 | 0.0797 | 0.3593 | BIO       | DiGress   | 0.2712 | 0.5202 | 0.1127 | 0.3188 |
> > |         | LGGM-X    | 0.4723 | 0.6843 | 0.2924 | 0.7555 |           | LGGM-X    | 0.1081 | 0.2696 | 0.0900 | 0.2053 |
> > | ASN     | DiGress   | 0.1496 | 0.3258 | 0.1506 | 0.4420 | ECON      | DiGress   | 0.2987 | 0.4841 | 0.2162 | 0.3834 |
> > |         | LGGM-X    | 0.0281 | 0.2440 | 0.0830 | 0.0618 |           | LGGM-X    | 0.1213 | 0.0920 | 0.1120 | 0.1086 |
> > | EMAIL   | DiGress   | 0.2192 | 0.6012 | 0.0702 | 0.3416 | RT        | DiGress   | 0.4164 | 0.1327 | 0.4147 | 0.5957 |
> > |         | LGGM-X    | 0.0751 | 0.2364 | 0.0768 | 0.3089 |           | LGGM-X    | 0.0525 | 0.1429 | 0.1330 | 0.2219 |
> > | WEB     | DiGress   | 0.2556 | 0.6186 | 0.1877 | 0.6045 | COL       | DiGress   | 0.2473 | 0.5826 | 0.2314 | 0.7679 |
> > |         | LGGM-X    | 0.0648 | 0.3961 | 0.0549 | 0.1127 |           | LGGM-X    | 0.0736 | 0.5769 | 0.0895 | 0.0988 |
> > | ROAD    | DiGress   | 0.3705 | 0.8226 | 0.2801 | 0.7198 | ECO       | DiGress   | 0.5431 | 0.7915 | 0.2338 | 0.6045 |
> > |         | LGGM-X    | 0.0713 | 0.2193 | 0.0987 | 0.2986 |           | LGGM-X    | 0.4753 | 0.3904 | 0.3194 | 0.3934 |
> > | POWER   | DiGress   | 0.3726 | 0.4582 | 0.3270 | 1.4732 | CITATION  | DiGress   | 0.2527 | 0.7790 | 0.1315 | 0.4966 |
> > |         | LGGM-X    | 0.0119 | 0.1293 | 0.0373 | 0.0754 |           | LGGM-X    | 0.1348 | 0.7257 | 0.1160 | 0.4981 |
> > | ALL     | DiGress   | 0.3112 | 0.5622 | 0.2030 | 0.5923 |           |           |        |        |        |        |
> > |         | LGGM-X    | 0.1408 | 0.3422 | 0.1253 | 0.2616 |           |           |        |        |        |        |
> >
> >
> >
> >
> > **Q3. In line 866 (Appendix C), the equation number is missing.**
> >
> > Thank you for the very sharp observation. We will correct the missing equation in the paper revision.

---

### Meta-Review · Area_Chair_e2xq · 2024-12-18

**Metareview:**

The paper introduces an approach to training graph generative models using a large corpus of graphs from multiple domains.

The strengths of this paper include its comprehensive coverage of various domains, the introduction of zero-shot and fine-tuning capabilities, and the innovative text-to-graph generation feature. However, the paper also has some weaknesses, such as performance variability across domains, limited training data size, technical contributions, and scalability concerns. The authors have partially addressed these issues by clarifying the novelty of their cross-domain pretraining approach, discussing potential scalability solutions, and providing additional experimental results.

In light of the authors’ responses and the overall contribution of the paper, I suggest a weak accept. Authors are encouraged to include some of the discussion in the revised paper.

**Additional Comments On Reviewer Discussion:**

Reviewers raised issues on performance variability across domains, limited training data size, technical contributions, and scalability concerns. The authors have partially addressed these issues by clarifying the novelty of their cross-domain pretraining approach, discussing potential scalability solutions, and providing additional experimental results.

---

### Decision · Program_Chairs · 2025-01-22

Accept (Poster)